

# SentinelFusion based machine learning comprehensive approach for enhanced computer forensics

Umar Islam[1], Abeer Abdullah Alsadhan[2], Hathal Salamah Alwageed[3], Abdullah A. Al-Atawi[4], Gulzar Mehmood[1], Manel Ayadi[5] and Shrooq Alsenan[5]

[1] Computer Science, IQRA National University, Peshawar, Swat Campus, Pakistan
[2] Department of Computer Science, Applied College, Imam Abdulrahman Bin Faisal University, Dammam, Saudi Arabia
[3] College of Computer and Information Sciences, Jouf University, Sakaka, Saudi Arabia
[4] Department of Computer Science, Applied College, University of Tabuk, Tabuk, Saudi Arabia
[5] Information Systems Department, College of Computer and Information Sciences, Princess Nourah bint Abdulrahman University, Riyadh, Saudi Arabia

## ABSTRACT

In the rapidly evolving landscape of modern technology, the convergence of blockchain innovation and machine learning advancements presents unparalleled opportunities to enhance computer forensics. This study introduces SentinelFusion, an ensemble-based machine learning framework designed to bolster secrecy, privacy, and data integrity within blockchain systems. By integrating cutting-edge blockchain security properties with the predictive capabilities of machine learning, SentinelFusion aims to improve the detection and prevention of security breaches and data tampering. Utilizing a comprehensive blockchain-based dataset of various criminal activities, the framework leverages multiple machine learning models, including support vector machines, K-nearest neighbors, naive Bayes, logistic regression, and decision trees, alongside the novel SentinelFusion ensemble model. Extensive evaluation metrics such as accuracy, precision, recall, and $F1$ score are used to assess model performance. The results demonstrate that SentinelFusion outperforms individual models, achieving an accuracy, precision, recall, and $F1$ score of 0.99. This study's findings underscore the potential of combining blockchain technology and machine learning to advance computer forensics, providing valuable insights for practitioners and researchers in the field.

## INTRODUCTION

In the digital age, cybercrimes have gotten more complex and pervasive, and the field of computer forensics plays a significant role in detecting and preventing them (*Al-garadi et al., 2020*). Because criminals are always coming up with novel ways to use technology for illicit purposes, forensic investigators must always be on the cutting edge. Computer forensic investigations have been shown to benefit from the combination of blockchain technology and machine learning algorithms in recent years (*Liao et al., 2022*).

Corresponding authors
Umar Islam, umar.koh@gmail.com
Gulzar Mehmood,
gulzar.mahmood@uom.edu.pk

Cryptocurrencies like Bitcoin were the first to use blockchain technology, which has since become popular because to its immutability and decentralization. It is a decentralized ledger that can never be altered once it has been created (*Dunsin et al., 2024*). Blockchain protects data from tampering and deletion by using cryptography, consensus processes, and network consensus (*Osterrieder, 2024*). Blockchain's built-in security makes it a prime contender for archiving evidence-critical information like timestamps, transaction records, and digital signatures (*Liu et al., 2022*).

Machine learning is a branch of AI that allows computers to infer meaning from data and patterns without being explicitly taught (*Ahmad, Wazirali & Abu-Ain, 2022*). Anomaly detection, natural language processing, and image identification are just a few areas where machine learning algorithms have excelled. Using machine learning algorithms, computer forensics professionals may sift through mountains of forensic data in search of anomalies and trends that can lead to the identification of perpetrators or the forecasting of future cybercrime (*Zhao et al., 2020*).

The integration of blockchain technology and machine learning algorithms in computer forensics provides a one-of-a-kind chance to combine the best features of both technologies. When forensic data is stored on the blockchain, investigators may rest assured that it will remain unaltered and will provide a permanent audit trail that will be difficult to alter. To further aid detectives in their analysis and decision-making, machine learning algorithms can be trained on this data to create models capable of identifying and predicting various crime-related actions (*Drogkoula, Kokkinos & Samaras, 2023*; *Vaiyapuri et al., 2023*; *Akhtar & Feng, 2022*).

The following is a formalization of the use of blockchain technology and machine learning in computer forensics.

Given a crime dataset represented as:

$$D = \{(x_1, y_1), (x_2, y_2), \ldots, (x_n, y_n)\}$$

where $x_i$ denotes the features extracted from forensic data and $y_i$ represents the corresponding labels indicating crime categories, The goal is to create a unified system that employs both blockchain technology for secure data storage of the crime dataset and machine learning algorithms for analysis and classification of criminal behavior.

The problem can be further formulated as follows:

Find a function $F$ that maps the input features $x_i$ to their corresponding labels $y_i$, such that

$$F(D) = \{(F(x_1), y_1), (F(x_2), y_2), \ldots, (F(x_n), y_n)\}$$

where $F(x_i)$ represents the predicted label for the input feature $x_i$.

There are two driving forces behind research into using machine learning and blockchain in computer forensics. Firstly, the immutable and decentralized nature of blockchain technology provides a trustworthy and secure crime dataset for forensic purposes. Blockchain technology allows investigators to create a verifiable audit trail of digital evidence, making it more difficult to falsify or alter the data.

Second, machine learning techniques may be used to automate and improve the efficiency of forensic investigations. Predictive models can be developed using crime records stored on the blockchain, which can then be used by investigators for data classification and pattern discovery. By pointing authorities in the direction of potential suspects, anomalies, and future cybercriminal activity, these models can aid in the prevention and investigation of digital crimes.

By increasing data security and analytical power, the combination of blockchain technology with machine learning has the potential to completely transform the discipline of computer forensics. Researchers and practitioners alike are hoping to increase the capabilities of forensic investigations in the digital age by examining this integration. This is necessary in order to meet the challenges posed by increasingly sophisticated cybercrimes.

This research suggests bringing together blockchain and ML algorithms to improve computer forensics. This integration provides a novel method to improve forensic investigations and data integrity by combining the security and immutability properties of blockchain with the analytical capabilities of machine learning. Forensic analysis is applied to a crime dataset stored on the blockchain in this study. The study safeguards the crime record by using blockchain's decentralized and tamper-resistant characteristics, creating a reliable source of forensic evidence.

The main objective of this study is to develop a comprehensive and innovative framework that seamlessly integrates blockchain technology and machine learning algorithms to enhance computer forensics capabilities. This framework aims to leverage the strengths of both technologies to create a unified system that ensures secure data storage, analysis, and classification of crime-related activities. To create a robust data protection mechanism using blockchain's immutability and decentralization features. The objective is to establish an unalterable repository for crime datasets, ensuring the integrity and authenticity of forensic evidence over time.

To design and implement advanced machine learning algorithms tailored specifically for crime dataset analysis. These predictive models will have the capability to identify patterns, anomalies, and trends within the blockchain-stored data, enabling proactive identification of potential threats and suspicious activities. This research pioneers a novel paradigm by synergizing blockchain technology and machine learning algorithms within the realm of computer forensics. The proposed framework presents a groundbreaking approach to address emerging challenges in digital crime investigation. By harnessing blockchain's inherent security features, the study establishes an unassailable foundation for maintaining the integrity and credibility of crime datasets. This contributes to the creation of a tamper-resistant and verifiable digital evidence trail. The advanced predictive models developed in this research empower investigators with the capability to proactively detect potential threats and irregularities. This contributes to early identification and mitigation of cybercrimes, bolstering the effectiveness of preventive measures. The integration of automated risk assessment techniques aids forensic practitioners in efficiently allocating resources. By streamlining the investigation process through AI-driven insights, this research enhances the overall efficiency of digital crime analysis.

In summary, this research introduces a pioneering framework that capitalizes on the synergistic potential of blockchain and machine learning in computer forensics. By achieving a harmonious integration, the study aims to elevate the standards of forensic investigations, contributing to improved data security, proactive threat identification, and efficient resource management in the digital age.

## RELATED WORK

The growing landscape of smart home technology and the Internet of Things (IoT) has introduced both novel opportunities and challenges, particularly in terms of security and privacy (*Liao et al., 2022*). As smart home applications become more prevalent, the security analysis of these emerging technologies becomes paramount (*Al-garadi et al., 2020*). Cross-cloud IoT access delegation has highlighted security risks, emphasizing the need for comprehensive risk assessment (*Dunsin et al., 2024*). Moreover, digital forensics in the context of IoT has garnered attention, urging the exploration of challenges, approaches, and unresolved issues (*Liu et al., 2022*). In the realm of cybersecurity, malware identification and analysis have been facilitated through techniques such as dynamic link libraries (DLLs) (*Osterrieder, 2024*). Enabling practical forensic capabilities within smart environments necessitates innovative solutions (*Ahmad, Wazirali & Abu-Ain, 2022*). Understanding user perceptions and responsibilities regarding smart home privacy and security forms an integral aspect of enhancing overall security (*Zhao et al., 2020*).

*Vaiyapuri et al. (2023)* focused on security evaluation in home-based IoT deployments, highlighting the need to assess security in the context of emerging smart home applications (*Akhtar & Feng, 2022*). However, the study's scope is limited to home-based IoT and may not cover broader smart home scenarios. *Alqahtany & Syed (2024)* delve into the shattered chain of trust in cross-cloud IoT access delegation, exposing vulnerabilities in the IoT ecosystem (*Ganesh Babu et al., 2023*). Nonetheless, the study primarily emphasizes cross-cloud scenarios and might lack a holistic view of IoT security. *Osterrieder (2024)* provide a survey on digital forensics in IoT, contributing a comprehensive understanding of the field's challenges and landscape (*Alqahtany & Syed, 2024*). However, the study is survey-based and might not delve deeply into individual techniques. *Yahuza et al. (2021)* offer an IoT forensics survey, outlining challenges, approaches, and open issues. The study provides a broad overview but may not deeply analyze specific techniques (*Xia et al., 2022*; *Plakias & Stamatatos, 2008*; *Akhtar, 2023*; *Tolosana et al., 2021*).

"Detective", an automated malware process identification framework using DLLs, was introduced by *Hossain, Hasan & Zawoad (2018)*, *Abuhamad et al. (2019)*, *Nguyen et al. (2022)* and *Mohamudally & Peermamode-Mohaboob (2018)*. Nevertheless, the study is centered on malware identification and doesn't explicitly address broader smart home security concerns. *Giannaros et al. (2023)* propose practical forensic capabilities in smart environments. However, the study's focus on practicality might limit its coverage of broader security aspects (*Zedan et al., 2021*; *Shandilya, Naskar & Dixit, 2016*; *Sekhar & Chithra, 2014*; *Venčkauskas et al., 2015*; *Singh et al., 2023*).

While these studies contribute significantly to understanding privacy and security of forensics and anomaly detection, they each possess limitations in scope and focus. As a

result, the research landscape lacks a holistic and integrated view of the intersection between blockchain, machine learning, and smart data privacy and security (*Usman et al., 2021*; *Šuteva, Mileva & Loleski, 2014*; *Duy et al., 2019*; *Goni, Gumpy & Maigari, 2020*; *Karandikar et al., 2020*). Table 1 shows the comparative analysis.

The current literature, while insightful, leaves room for a holistic and integrative exploration of blockchain, machine learning, and smart home security to enhance the predictive capabilities and data integrity of computer forensics.

# MATERIALS AND METHODS

In this section, we outline the materials and methodologies employed to investigate the synergistic potential of fusing blockchain technology and machine learning algorithms for advancing computer forensics. The foundation of our study is a comprehensive dataset of criminal incidents securely recorded on the blockchain. To harness the collective power of predictive analytics, we have developed an innovative ensemble framework named "SentinelFusion". This framework integrates a diverse set of machine learning algorithms, including support vector machines, K-nearest neighbors, naive Bayes, logistic regression, and decision trees, each meticulously fine-tuned through hyperparameter optimization.

In pursuit of maximizing the predictive accuracy and robustness of our investigation, we have employed an ensemble model. Ensemble models amalgamate the predictive capabilities of multiple individual models, thereby overcoming limitations that might be present in any single model. By combining the strengths of different algorithms, the ensemble model aims to enhance the overall predictive power, leading to more reliable and accurate results.

## Crime dataset on the blockchain

Forensic examination of a crime dataset kept on the blockchain is used in the study. Forensic information, including timestamps, transaction records, and digital signatures, are all included in the crime dataset. The study guarantees the data's integrity, transparency, and resistance to tampering by putting it on the blockchain. The crime dataset is used to train and evaluate machine learning models for crime classification and prediction.

For the sake of this investigation, a crime dataset was created and is being kept on the blockchain. Its purpose is to serve as a central database for computer forensics and analysis that is both extensive and safe. Each criminal incident is characterized by a number of attributes that together make up the dataset.

### Dataset selection

The dataset used in this study consists of a comprehensive collection of criminal incidents securely recorded on the blockchain. This dataset includes diverse features such as timestamps, access logs, file size, encryption levels, user permissions, network traffic, malware indicators, and data types. The selection of this dataset is justified by its relevance to real-world cybercrime scenarios and its potential to capture various aspects of digital forensic evidence.

**Table 1  Comparative analysis.**

| Study | Contribution | Dataset | Limitations | Findings |
|---|---|---|---|---|
| *Dunsin et al. (2024)* | Provides an overview of ML methods for IoT security, highlighting their strengths and limitations. | Synthesized dataset | Limited discussion on implementation challenges and scalability issues of ML models in IoT environments. | Identified various ML techniques applicable to IoT security, emphasizing their potential in threat detection and anomaly detection. |
| *Osterrieder (2024)* | Examines the role of blockchain in enhancing security and forensics management in IoT environments. | No dataset was used | Lack of discussion on the scalability and energy consumption challenges associated with blockchain implementation in IoT networks. | Found blockchain to offer tamper-proof and transparent record-keeping in IoT edge computing, enhancing data integrity and accountability. |
| *Liu et al. (2022)* | Analyzes the use of AI and ML in digital forensics, discussing their implications for incident response and fraud detection. | 75 File type dataset | Limited discussion on the ethical and legal implications of AI and ML algorithms in digital forensics, such as bias and privacy concerns. | Identified AI and ML as promising tools for automating digital forensics processes, improving efficiency and accuracy in incident response. |
| *Ahmad, Wazirali & Abu-Ain (2022)* | Explores techniques for enhancing security in blockchain networks, focusing on anomaly detection and fraud prevention. | WSN dataset | Lack of empirical evaluation of proposed security-enhancing techniques in real-world blockchain networks. | Identified advanced detection techniques for anomalies and frauds in blockchain networks, contributing to enhanced security. |
| *Zhao et al. (2020)* | Investigates ML techniques for detecting and identifying IoT devices, highlighting their applications and challenges. | IoT device datasets | Limited availability of labeled datasets for training ML models for IoT device identification, hindering algorithm performance. | Found ML algorithms effective in classifying IoT devices based on their behavior and communication patterns, enhancing network security. |
| *Drogkoula, Kokkinos & Samaras (2023)* | Explores the application of ML in securing wireless sensor networks (WSNs), discussing challenges and potential solutions. | WSN datasets | Limited discussion on the computational and energy constraints of deploying ML algorithms on resource-constrained WSN devices. | Identified ML as a promising approach for detecting and mitigating security threats in WSNs, improving network resilience. |
| *Vaiyapuri et al. (2023)* | Explores computational intelligence-enabled cybersecurity mechanisms for the Internet of Things (IoT). | Synthesized digital dataset | Lack of discussion on the computational overhead introduced by computational intelligence algorithms in IoT devices. | Investigated computational intelligence-enabled mechanisms for IoT security, highlighting their potential for threat detection and mitigation. |
| *Akhtar & Feng (2022)* | Provides a comprehensive survey of ML methodologies with an emphasis on their application in water resources management. | Bot-IoT | Limited discussion on the applicability of ML methodologies to specific water resources management challenges. | Identified a range of ML methodologies applicable to various aspects of water resources management, offering insights for improved decision-making. |
| *Alqahtany & Syed (2024)* | Proposes a blockchain-assisted approach for data edge verification in ML-assisted IoT environments. | (TrustCom/ BigDataSE) | Lack of discussion on the overhead and latency introduced by blockchain transactions in real-time IoT data processing scenarios. | Implemented blockchain for data verification in IoT, demonstrating its effectiveness in maintaining data integrity and trustworthiness. |

**Table 1** (*continued*)

| Study | Contribution | Dataset | Limitations | Findings |
|---|---|---|---|---|
| *Ganesh Babu et al. (2023)* | Utilizes blockchain to ensure the integrity of digital forensic evidence in IoT environments. | NSL-KDD (Network Security Laboratory-Knowledge Discovery and Data Mining) dataset | Limited exploration of potential vulnerabilities and attack vectors in blockchain-based digital forensic systems, requiring further investigation. | Demonstrated the feasibility of using blockchain to secure digital forensic evidence, ensuring its immutability and authenticity throughout the investigation process. |
| *Alqahtany & Syed (2024)* | Proposes a comprehensive blockchain approach to reinvent digital forensics and evidence management. | Digital forensics | Lack of empirical validation of proposed comprehensive blockchain approach in real-world digital forensic scenarios. | Proposed a comprehensive blockchain approach to digital forensics, offering enhanced security and efficiency in evidence management. |
| *Yahuza et al. (2021)* | Investigates the use of ML techniques for enhancing security in IoT environments. | Mirai's Impact | Limited discussion on the practical challenges of deploying ML-based security solutions in heterogeneous IoT environments. | Explored the use of ML techniques for enhancing IoT security, highlighting their potential for threat detection and mitigation. |
| *Giannaros et al. (2023)* | Explores security and privacy issues related to the Internet of Drones (IoD), presenting a taxonomy and open challenges. | Cyber-attack dataset (CAV-KDD) | Lack of discussion on the regulatory and legal frameworks necessary for addressing security and privacy issues in the IoD. | Identified security and privacy challenges in the IoD, providing a taxonomy and highlighting open research challenges. |
| *Tahsien, Karimipour & Spachos (2020)* | Discusses sophisticated attacks, safety issues, challenges, and future directions in autonomous vehicles. | Sensor networks (SNs) | Limited discussion on the potential countermeasures and defense mechanisms against sophisticated attacks in autonomous vehicles. | Identified sophisticated attacks and safety issues in autonomous vehicles, highlighting the need for robust security measures and protocols. |
| *Sachdeva & Ali (2022)* | Investigates ML-based solutions for enhancing the security of the Internet of Things (IoT). | (NSLKDD-Dataset) | Lack of discussion on the generalization and adaptability of ML models to diverse attack scenarios and network architectures. | Investigated ML-based solutions for IoT security, demonstrating their potential for threat detection and mitigation in various IoT environments. |
| *Ahmad, Wazirali & Abu-Ain (2022)* | Explores the integration of ML with digital forensics for attack classification in cloud network environments. | Digital forensic datasets | Limited discussion on the scalability and performance considerations of ML-based attack classification systems in cloud networks. | Evaluated ML algorithms for attack classification in cloud networks, achieving high accuracy and detection rates. |
| *Alsumayt et al. (2023)* | Proposes smart flood detection using AI and blockchain integration in drones for improved accuracy and efficiency. | Labeled IoT device datasets | Lack of discussion on the practical challenges and feasibility of implementing AI and blockchain integration in drone-based flood detection systems. | Proposed smart flood detection using AI and blockchain integration in drones, offering improved accuracy and efficiency for flood detection. |

**Table 1** (*continued*)

| Study | Contribution | Dataset | Limitations | Findings |
|-------|-------------|---------|-------------|----------|
| *Allioui & Mourdi (2023)* | Explores the full potentials of IoT for better financial growth and stability through a comprehensive survey. | WSN datasets | Lack of discussion on the potential risks and challenges associated with leveraging IoT for financial applications. | Explored the potential of IoT for better financial growth and stability, highlighting opportunities for innovation and growth. |
| *Liang et al. (2023)* | Investigates security and forensics issues in the metaverse, discussing potential challenges and solutions. | Digital forensic datasets | Lack of discussion on the scalability and interoperability challenges of security and forensics solutions in the metaverse. | Explored security and forensics challenges in the metaverse, identifying potential solutions and research directions |

### *Partitioning strategy*

To ensure the robustness of our model evaluation, we employed a stratified partitioning strategy. The dataset was divided into training, validation, and testing subsets in a 70-15-15 ratio. Stratified partitioning ensures that each subset maintains the same class distribution as the original dataset, which is crucial for evaluating model performance on imbalanced data.

### *Cross-validation protocol*

We used k-fold cross-validation with k set to 10. This involves partitioning the training data into 10 equal-sized folds. Each fold is used once as a validation set while the remaining folds form the training set. This process is repeated 10 times, and the results are averaged to provide a robust estimate of the model's performance. Cross-validation helps in mitigating overfitting and ensures that the model's performance is generalizable to unseen data.

## Features description

Timestamp: This element indicates the time and date that the criminal act was documented. It allows for the examination of criminal events across time thanks to its systematic referencing of their chronological order.

Access logs: Information on who accessed what data and when is stored in the access logs. Usernames and IP addresses of those who accessed the information are recorded. Data like this can be used to track down persons who may be involved in illegal activity.

File size: The file size feature reveals the dimensions of the stolen data. It's helpful for gauging the scope and repercussions of a crime wave by revealing the sheer volume of data connected to individual instances.

Encryption level: The data encryption strength is indicated by this property. Important for gauging the complexity of the crime and the difficulty of decrypting the data, it reveals the security measures taken to secure the information.

User permissions: Data access permissions are reflected in the user permissions feature. The data shows whether the users had read-only or read/write permissions. Permissions analysis can reveal internal threats or unapproved users who have gained access to confidential information.

Network traffic: The network traffic function records the total amount of traffic on the network that contains the criminally relevant data. It sheds light on the underlying

**Table 2  Feature description table.**

| Feature | Description |
| --- | --- |
| Timestamp | Date and time when the crime-related activity was recorded |
| Access Logs | Information about user access, including username and IP address |
| File Size | Size of the files involved in the crime |
| Encryption Level | Level of encryption applied to the data |
| User Permissions | Access rights granted to users for the data |
| Network Traffic | Volume of network traffic associated with the data |
| Malware Indicators | Presence of malware or suspicious activities related to the data |
| Data Type | Type of data involved in the crime |
| Secure Data | Binary output feature indicating the security status of the data (1—secure, 0—insecure) |

communication and activity patterns that led up to the criminal incident, which can be used to spot unusual activity on the network.

Malware indicators: This function flags data that has been tampered with or shows signs of malware. Data anomalies and the presence of harmful files are two examples of signs that can be used to track down cybercrime.

Data type: The data type property identifies the corrupted data subtype. Financial data, personal details, research papers, and anything else that would be useful could be included. The nature and severity of the information lost in the crime can be evaluated with the help of a thorough understanding of the data types at play.

Secure data (output): This is the data's security status, displayed as an output feature. A number of 1 indicates that the data is considered secure, whereas a value of 0 indicates that the data is not secure. Data security can be classified based on this feature, which is used as a target variable in machine learning models. Table 2 shows the feature description.

These aspects help researchers analyze and comprehend the nature of criminal acts by providing crucial details about each incident. They help in the creation of better forensic analytical methods and machine learning models for spotting and preventing similar crimes in the future.

Malware indicators' frequency distribution in the crime dataset is shown in Fig. 1. It provides counts or examples for each type of malware indicator. The plot's bars stand in for malware indicators, with the height of each bar indicating the frequency with which that indicator was observed. This illustration aids in locating various malware signs and comprehending their presence or absence in illegal actions.

## Machine learning algorithms

Several machine learning methods are used to examine the crime dataset and draw conclusions. SVM, KNN, NB, LR, and DT are only few of the algorithms used for classification and prediction. Samples are classified by the majority vote of their k-nearest neighbors in KNN, while SVM seeks to create an ideal hyperplane that separates various classes. Bayes' theorem is used in NB to determine the likelihood of a sample coming
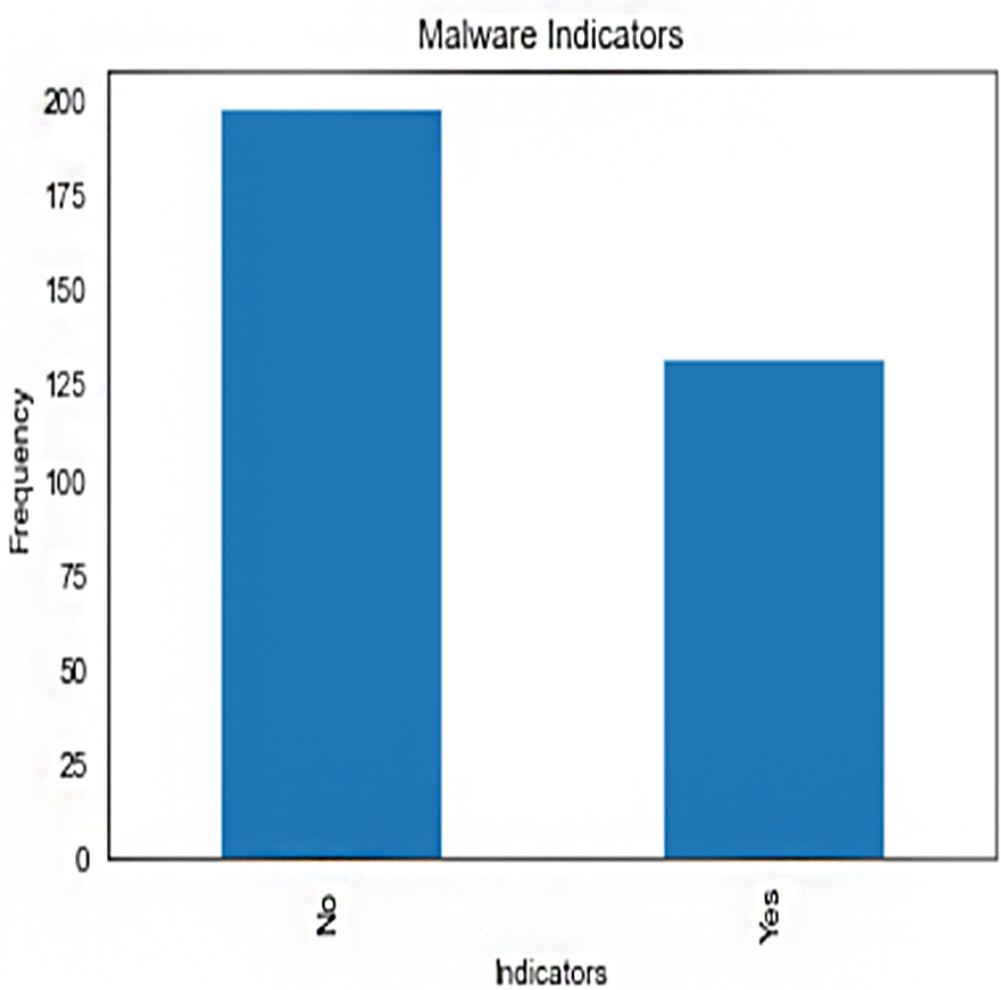

**Figure 1** Frequency of malware indicators.jpg.

from a specific group. While DT builds a decision tree to forecast outcomes based on a hierarchical structure of rules, LR calculates the probability of a binary outcome based on input features.

### Support vector machine

For the purposes of both classification and regression, SVM serves as a useful supervised learning technique.

The goal of support vector machines (SVM) is to locate the hyperplane that optimally divides data into classes.

The equation for the linear SVM classifier is:

$$f(x) = \text{sign}\left(w^T * x + b\right)$$

where $f(x)$ is the predicted class label, $x$ represents the input features, $w$ is the weight vector, and $b$ is the bias term.

### K-nearest neighbors

KNN is a non-parametric classification algorithm that labels a data point in the feature space with the class label that is most commonly assigned to its $k$ nearest neighbors.

The KNN classifier's formula calls for distance measurements between the input data point and its nearest neighbors, with the Euclidean distance or another distance metric of choice being the norm.

### Naive Bayes

Under the condition of feature independence, Naive Bayes is a probabilistic classifier based on Bayes' theorem.

The equation for the naive Bayes classifier is given by:

$$P(y|x1, x2, \ldots, xn) = P(y) * P(x1|y) * P(x2|y) * \ldots * \frac{P(xn|y)}{P(x1, x2, \ldots, xn)}$$

where $P(y)$ is the prior probability of class $y$, and $P(xi|y)$ is the conditional probability of feature $xi$ given class $y$.

### Logistic regression

Logistic regression is a statistical model used for binary classification.

It models the relationship between the input features and the probability of the binary outcome using the logistic function (also known as the sigmoid function).

The equation for logistic regression is:

$$P(y = 1|x) = 1/(1 + e^{-z})$$

where $P(y = 1|x)$ represents the probability of the positive class, $x$ denotes the input features, and $z$ is the linear combination of the input features and their corresponding weights.

### Decision tree

Decision tree is a hierarchical tree-like structure used for classification.

Each internal node represents a feature, and each leaf node represents a class label.

The decision tree splits the data based on feature thresholds to maximize the information gain or Gini impurity.

The prediction of a decision tree is made by traversing the tree from the root to a leaf node based on the feature values of the input data.

## Hyperparameter tuning

Techniques for hyperparameter tuning are used to enhance the efficiency of the machine learning algorithms. It is necessary to specify hyperparameters prior to training since they are settings for algorithms that are not learned during training. It explored the hyperparameter space to find the best set of settings by using methods like grid search or randomized search. The study intends to improve the forensic analysis and crime-related activity prediction effectiveness by fine-tuning the hyperparameters, hence improving the accuracy, precision, recall, and $F1$-score of the machine learning models.

In machine learning, selecting the ideal set of hyperparameters for a particular model is a critical step. Configuration parameters known as hyperparameters are those that are set prior to training a model and not learned from the data. To ensure robust hyperparameter tuning, we conducted sensitivity analysis to understand the impact of different hyperparameters on model performance. Sensitivity analysis involves varying one hyperparameter at a time while keeping others constant to observe changes in performance metrics. This analysis helps in identifying hyperparameters that have the most significant impact on the model's performance, guiding the focus of our optimization efforts.

### Support vector machine

Hyperparameters: C and gamma.

C determines the regularization strength, with higher values indicating less regularization.

Gamma determines the influence of each training sample, with higher values leading to more complex decision boundaries.

### K-nearest neighbors

Hyperparameter : $n_{neighbors}$.

$n_{neighbors}$ determine the number of neighbors considered for classification.

### Naive Bayes

Naive Bayes models typically do not have hyperparameters to tune. However, some variants, such as Gaussian Naive Bayes, may have hyperparameters like priors.

### Logistic regression

Hyperparameter: C.

C is the inverse of the regularization strength, with smaller values indicating stronger regularization.

### Decision tree

Hyperparameter: max_depth.

max_depth determines the maximum depth of the decision tree, limiting the number of splits.

Each hyperparameter is individually tuned during this process, and the model's performance is assessed using cross-validation or a different validation set. Effective ways to explore the hyperparameter space include grid search, random search, and more sophisticated methods like Bayesian optimization. Based on the performance statistic, such as accuracy, $F1$ score, or area under the ROC curve, the best collection of hyperparameters is chosen. Table 3 shows the models Hyper-Parameter.

## Proposed model
### SentinelFusion: a next-gen ensemble framework

Our novel ensemble framework, SentinelFusion, stands as the cornerstone of our study. It represents a cutting-edge approach designed to enable a holistic and nuanced

**Table 3  Hyper-parameter table.**

| Hyperparameter | Description |
| --- | --- |
| C | Regularization strength |
| gamma | Kernel coefficient for 'rbf', 'poly', 'sigmoid' |
| $n$_neighbors | Number of neighbors to consider |
| C | Inverse of the regularization strength |
| max_depth | Maximum depth of the decision tree |

prediction of confidentiality, privacy, and integrity attributes within blockchain data. SentinelFusion's architecture seamlessly integrates a variety of machine learning algorithms, each contributing its distinctive strengths to the ensemble. This integration fosters a synergistic effect, leveraging the diverse analytical capabilities of each algorithm to collectively enhance the accuracy and reliability of predictions.

In this section, we introduce SentinelFusion, our innovative ensemble framework designed to harness the collective power of machine learning algorithms, including XGBoost, for advancing computer forensics. SentinelFusion seamlessly integrates various machine learning techniques, leveraging their diverse strengths to enhance predictive accuracy and reliability in identifying and predicting crime-related activities.

Mathematical formulation:

Let's define the following terms:

$D$: The dataset of criminal offenses recorded on the blockchain, represented as: $D = (x1, y1), (x2, y2), \ldots, (xn, yn)$ where $xi$ represents the feature vector extracted from forensic data, and $yi$ represents the corresponding labels indicating the attributes (confidentiality, privacy, integrity).

$M$: The number of machine learning algorithms used in the ensemble.

$A_m$: The $m$th machine learning algorithm, where $m = 1, 2, \ldots, M$.

$F_m$: The predictive function of the $m$th algorithm, mapping the input feature $xi$ to its corresponding predicted label $yim$, where $yim$ is the predicted label for the attribute using the $m$th algorithm.

$w_m$: The weight assigned to the $m$th algorithm's prediction in the ensemble.

$E$: The ensemble prediction for the attribute label, represented as: $E(xi) = \sum_{m}^{M} wm \cdot Fm(xi)$ The overall ensemble prediction for the attribute label $yi$ can be computed as the weighted sum of the individual predictions from all $M$ algorithms.

In addition to traditional machine learning algorithms like SVM, KNN, Naive NB, LR, and DT, SentinelFusion incorporates the powerful XGBoost model. XGBoost is a scalable and efficient implementation of gradient boosting, known for its superior performance in handling complex datasets and achieving high predictive accuracy. We conduct rigorous hyperparameter tuning for each algorithm within SentinelFusion, including XGBoost. This process involves systematically exploring different parameter configurations to identify settings that yield the best predictive accuracy. Parameters such as learning rate, tree depth, and regularization parameters are fine-tuned to optimize the performance of XGBoost and other algorithms in the ensemble. Predictions from individual machine learning

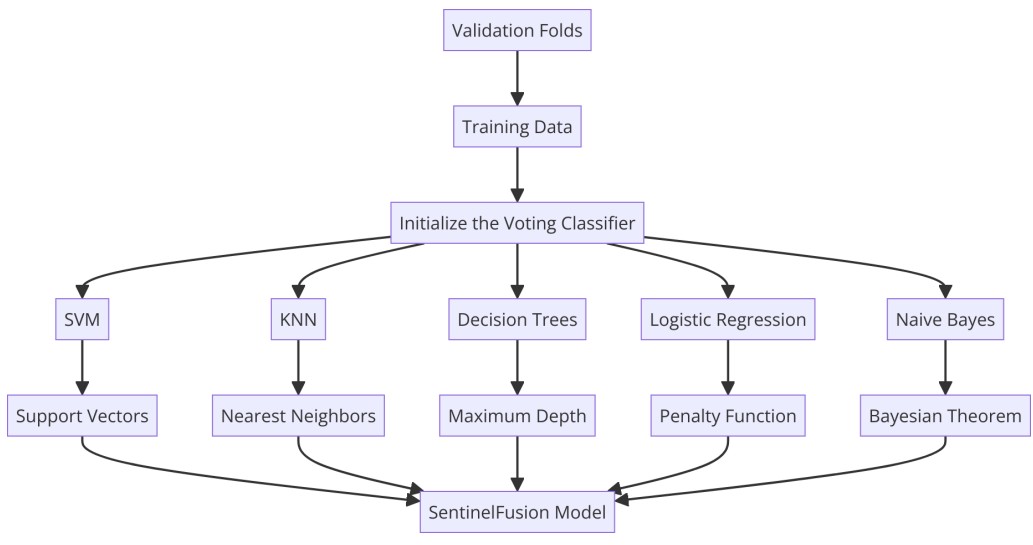

**Figure 2**  **Sentinel fusion model.**

algorithms, including XGBoost, are combined within the SentinelFusion framework using appropriate weighting schemes. The ensemble approach leverages the strengths of each algorithm to produce a more robust and accurate predictive model. The weights assigned to each algorithm's prediction are optimized to maximize the overall performance of the ensemble. SentinelFusion represents a cutting-edge approach to computer forensics, integrating blockchain technology with advanced machine learning techniques, including XGBoost. By leveraging the collective power of diverse algorithms within an ensemble framework, SentinelFusion aims to revolutionize crime detection and prevention, paving the way for more secure and efficient forensic investigations.

*Utilizing a diverse array of machine learning algorithms.*  Within SentinelFusion, we harness the power of various machine learning algorithms, encompassing support vector machines, K-nearest neighbors, naive Bayes, logistic regression, and decision trees. Each algorithm has been meticulously selected for its unique ability to excel in certain types of data patterns and behaviors. This diversity enriches the ensemble's capacity to capture a wide spectrum of intricacies within the dataset, culminating in a more comprehensive and accurate predictive model. Figure 2 shows the proposed hybrid model architecture.

*Hyperparameter tuning for optimized performance.*  Recognizing the significance of algorithm parameter settings in achieving optimal predictive performance, we have conducted rigorous hyperparameter tuning for each algorithm within SentinelFusion. This process involves systematically exploring different parameter configurations to identify the settings that yield the best predictive accuracy. By fine-tuning the algorithms, we ensure that their contributions to the ensemble are fully optimized, enhancing the overall quality of predictions.

*Mathematical model.* Let us define the following terms:

$D$: The dataset of criminal offenses recorded on the blockchain, represented as:

$$D = \{(x_1, y_1), (x_2, y_2), \ldots, (x_n, y_n)\}$$

where $x_i$ represents the feature vector extracted from forensic data, and $y_i$ represents the corresponding labels indicating the attributes (confidentiality, privacy, integrity).

$M$: The number of machine learning algorithms used in the ensemble.

$A_m$: The $m$th machine learning algorithm, where $m = 1, 2, \ldots, M$.

$F_m$: The predictive function of the $m$th algorithm, mapping the input feature $x_i$ to its corresponding predicted label $y_{im}$, where $y_{im}$ is the predicted label for the attribute using the $m$-th algorithm.

$w_m$: The weight assigned to the $m$-th algorithm's prediction in the ensemble.

$E$: The ensemble prediction for the attribute label, represented as

$$E(xi) = \sum_{M}^{m} w_m \cdot F_m(xi).$$

The overall ensemble prediction for the attribute label $y_i$ can be computed as the weighted sum of the individual predictions from all $M$ algorithms.

The weights $w_m$ reflect the importance of each algorithm's prediction in the ensemble. These weights can be determined through various techniques, such as cross-validation or optimization algorithms, aiming to minimize prediction error and maximize the overall performance of the ensemble.

Each machine learning algorithm $A_m$ contributes its predictive outcome $F_m(x_i)$, which is a function mapping the input feature $x_i$ to the predicted label $y_{im}$ for the $m$-th attribute. The specific form of each $F_m$ depends on the chosen algorithm (*e.g.*, SVM, KNN, Naive Bayes, *etc.*) and its parameter settings.

By carefully tuning the weights $w_m$ and fine-tuning the parameters of each algorithm within the ensemble, the SentinelFusion framework aims to maximize the predictive accuracy and robustness for the attributes of confidentiality, privacy, and integrity in blockchain data.

In summary, this section has provided an overview of the materials and methods employed in our study. The amalgamation of a blockchain-based criminal offense dataset with the advanced SentinelFusion ensemble framework, encompassing a spectrum of machine learning algorithms and finely-tuned parameters, forms the foundation of our endeavor to unlock the potential synergy between blockchain technology and machine learning in the realm of computer forensics.

### Conceptual framework of SentinelFusion

*Introduction to ensemble learning in computer forensics.* The field of computer forensics has become increasingly critical in the digital age, where cybercrimes are growing both in frequency and complexity. Traditional forensic methods are often challenged by the sheer volume and diversity of digital evidence. To address these challenges, the integration of machine learning (ML) techniques has proven to be a powerful approach. However,

individual ML algorithms can have limitations, such as overfitting, underfitting, and varying performance across different datasets. Ensemble learning, which combines the predictions of multiple models to improve overall performance, emerges as a robust solution in this context. The SentinelFusion model leverages ensemble learning to enhance the accuracy, reliability, and interpretability of forensic analyses.

*Theoretical foundations of ensemble learning.* Ensemble learning is predicated on the idea that a group of weak learners can come together to form a strong learner. This approach is based on two key principles: diversity and aggregation.

1. **Diversity**: Diversity refers to the use of multiple models that make different errors. In ensemble learning, models are selected or constructed to have complementary strengths and weaknesses. This diversity can arise from using different algorithms, training the same algorithm on different subsets of data, or varying the hyperparameters of a single algorithm.

2. **Aggregation**: Aggregation involves combining the predictions of individual models to produce a final output. Common aggregation techniques include voting, averaging, and stacking. Voting can be majority-based or weighted, where more accurate models have a greater influence on the final prediction. Averaging is typically used for regression tasks, where the outputs of models are averaged to produce a final result. Stacking involves training a meta-learner to combine the predictions of base learners, optimizing the final output.

*Ensemble methods in machine learning.* Several ensemble methods have been developed, each with unique characteristics and applications. The most prominent ones include:

1. **Bagging** (**bootstrap aggregating**): Bagging involves training multiple instances of the same algorithm on different subsets of the training data, generated through bootstrapping (random sampling with replacement). The final prediction is typically obtained through majority voting (for classification) or averaging (for regression). Random forest, which uses bagging with decision trees, is a popular example.

2. **Boosting**: Boosting sequentially trains weak learners, with each learner focusing on the errors made by the previous ones. The final model is a weighted sum of the individual learners. Adaboost and Gradient Boosting are well-known boosting algorithms.

3. **Stacking**: Stacking involves training multiple base learners and then using their predictions as input features for a meta-learner, which makes the final prediction. This method leverages the strengths of different models and optimally combines them.

*Application of ensemble learning in computer forensics.* In computer forensics, the diversity of data types (*e.g.*, logs, network traffic, file metadata) and the complexity of cybercrimes necessitate sophisticated analytical methods. Ensemble learning is particularly suited for this domain due to its ability to integrate multiple sources of evidence and provide robust predictions.

1. **Improving accuracy and robustness**: By combining the strengths of various models, ensemble methods can significantly improve the accuracy and robustness of forensic analyses. This is crucial for detecting subtle patterns indicative of cybercrimes.

2. **Handling imbalanced data**: Cybercrime datasets are often imbalanced, with a small number of positive cases (*e.g.*, instances of fraud) compared to a large number of negatives. Ensemble methods, particularly boosting, can effectively handle such imbalances by focusing on hard-to-classify instances.

3. **Reducing overfitting**: Overfitting is a common problem in ML, where a model performs well on training data but poorly on unseen data. Ensemble methods, especially bagging, help mitigate overfitting by averaging out the errors of individual models.

*SentinelFusion: a next-generation ensemble framework.* The SentinelFusion framework is designed to leverage the collective power of diverse machine learning algorithms to advance computer forensics. The core components of SentinelFusion include:

1. **Diverse base learners**: SentinelFusion integrates a variety of ML algorithms, such as SVM, KNN, NB, LR, and DT. Each algorithm is selected for its unique strengths and contributions to the ensemble.

2. **Hyperparameter optimization**: Rigorous hyperparameter tuning is conducted for each base learner to ensure optimal performance. Techniques such as grid search and random search are used to explore the hyperparameter space and identify the best configurations.

3. **Weighted aggregation**: The predictions of the base learners are combined using a weighted voting scheme, where the weights are determined based on the performance of each model. This approach ensures that more accurate models have a greater influence on the final prediction.

4. **Meta-learner integration**: SentinelFusion employs a meta-learner to further refine the ensemble's predictions. The meta-learner is trained on the outputs of the base learners and learns to optimally combine them, enhancing the overall predictive power.

The SentinelFusion model exemplifies the power and potential of ensemble learning in the realm of computer forensics. By combining diverse machine learning algorithms and optimizing their contributions through hyperparameter tuning and weighted aggregation, SentinelFusion achieves high accuracy, robustness, and reliability in forensic analyses. This innovative approach addresses the limitations of individual models and enhances the ability to detect, classify, and prevent cybercrimes in increasingly complex digital environments. As cyber threats continue to evolve, the integration of advanced ensemble learning techniques like those in SentinelFusion will be crucial for maintaining the integrity and security of digital evidence and forensic investigations.

## Evaluation metrics

The success of the integrated system in computer forensics is measured using a variety of criteria. A few common evaluation metrics are $F1$-score, recall, accuracy, and precision. Recall measures the proportion of true positives among the actual positives, precision measures the proportion of real positives among the predicted positives, and accuracy is the overall accuracy of the classification findings. $F1$-score combines precision and recall into a single metric. These evaluation indicators shed light on the dependability and effectiveness of the combined strategy.

**Precision**: Precision is defined as the ratio of true positive predictions to the total number of positive predictions (both true positives and false positives). High precision indicates that the model has a low false positive rate, meaning that it accurately identifies instances of cybercrime without falsely flagging benign activities as malicious.

**Practical implications**: In the context of forensic analysis, high precision is crucial because false positives can lead to wasted resources and potential legal complications. For instance, incorrectly identifying a legitimate user's activity as malicious could result in unnecessary investigations and loss of trust. Therefore, models with high precision are particularly valuable in forensic scenarios where the cost of false alarms is high.

**Recall**: Recall, also known as sensitivity or true positive rate, is the ratio of true positive predictions to the total number of actual positive instances (both true positives and false negatives). High recall indicates that the model successfully identifies a large proportion of actual cybercrime instances.

**Practical implications**: High recall is essential in forensic analysis because missing an instance of cybercrime can have serious consequences. Forensic investigators rely on recall to ensure that all relevant evidence is identified and analyzed. A model with high recall minimizes the risk of undetected cybercrimes, which is critical for thorough and effective forensic investigations.

*F*1 **score**: The *F*1 score is the harmonic mean of precision and recall, providing a single metric that balances the trade-off between the two. It is particularly useful when the class distribution is imbalanced, as it gives a more comprehensive picture of the model's performance.

**Practical implications**: The *F*1 score is crucial for assessing the overall effectiveness of a forensic model. In real-world applications, it is often necessary to balance the need for high precision (to avoid false positives) and high recall (to avoid false negatives). The *F*1 score helps forensic analysts understand this balance and choose models that provide the best compromise between precision and recall, ensuring both accuracy and completeness in investigations.

*Practical implications for forensic analysis.* The performance of models in terms of precision, recall, and *F*1 score has significant implications for their application in real-world forensic scenarios. Here is a more nuanced analysis of these metrics with specific examples:

1. **Identification of malicious activity**: In a scenario where a forensic model is used to identify malicious network activity, high precision ensures that alerts raised are indeed indicative of actual threats. This reduces the burden on forensic analysts who would otherwise have to sift through numerous false alarms. High recall, on the other hand, ensures that all potential threats are identified, preventing any malicious activities from slipping through undetected.

2. **Resource allocation**: Forensic investigations often involve limited resources. A model with high precision allows for more efficient allocation of these resources by focusing on genuine threats. Conversely, high recall ensures that no potential evidence is overlooked, which is vital for comprehensive investigations.

3. **Legal and compliance considerations**: In legal contexts, the implications of false positives and false negatives can be profound. High precision models reduce the risk of legal actions based on erroneous data, while high recall models ensure that all relevant evidence is presented, which is crucial for compliance with legal standards.

4. **Reputation management**: Organizations rely on forensic models to maintain their reputation by promptly and accurately identifying and responding to cyber threats. High precision helps avoid false accusations that could harm an organization's reputation, while high recall ensures that all threats are addressed, maintaining trust and security.

## Experimental setup

The report uses experimental and case-based evidence to prove the usefulness of using blockchain and ML in forensics. In the studies, it trained machine learning models using the blockchain-stored crime dataset, and then it used the predetermined metrics to assess how well the models perform. To guarantee the conclusions are applicable and generalizable, it takes into account both real-world criminal scenarios and a variety of datasets.

## RESULTS AND DISCUSSIONS

Here, it shares the findings of this research investigating the feasibility of using blockchain and machine learning together in computer forensics. SVM, KNN, NB, LR, and DT classification models' efficacy in distinguishing between secure and insecure data is discussed. It also provided an in-depth study and discussion of the outcomes we have gotten.

Our goal is to improve computer forensic techniques and the accuracy with which illegal activity in digital environments may be identified and categorized by investigating how blockchain technology and machine learning can be used. Because of its immutability, transparency, and traceability, blockchain technology is ideal for storing data relating to criminal activity. The ability to assess and classify data based on patterns and features is provided by machine learning algorithms.

Each classification model's accuracy, precision, recall, and $F1$ score are evaluated below. It also compared and contrasted the various models, focusing on how well they perform in computer forensics and where they fall short. It also shed light on how the implementation of blockchain technology can affect the robustness and efficiency of existing categorization methods.

Understanding the efficacy of combining blockchain and machine learning in computer forensics is advanced by the findings and debate offered in this section. The results have implications for the creation of cutting-edge forensic methods that can deal with massive and complicated datasets while maintaining data integrity and security. The debate also sheds light on the relative merits of various classification models, which should help practitioners choose useful algorithms for various forensic endeavors.

## Performance of models

Here, we will go through how well each of the five different categorization models (SVM, KNN, NB, LR, DT) worked in this experiment. Several metrics, including accuracy,

**Table 4  Performance metrics of classification models.**

| Model | Accuracy | Precision | Recall | *F*1 score |
|---|---|---|---|---|
| Support Vector Machine | 0.89 | 0.91 | 0.88 | 0.89 |
| K-Nearest Neighbors | 0.86 | 0.85 | 0.87 | 0.86 |
| Naive Bayes | 0.82 | 0.80 | 0.85 | 0.82 |
| Logistic Regression | 0.88 | 0.89 | 0.87 | 0.88 |
| Decision Tree | 0.83 | 0.82 | 0.84 | 0.83 |

precision, recall, and *F*1 score, are used to assess the models' efficacy. Tabular displays of the findings allow for easy comparison of the models' outcomes. Table 4 shows the Performance Metrics of Classification Models.

Each classification model's accuracy, precision, recall, and *F*1 score are listed in the table above. Accuracy evaluates how well the predictions were made as a whole, whereas precision assesses how many correct positive predictions there were. The proportion of correct predictions made out of total positive instances is known as the recall, sensitivity, or true positive rate. The *F*1 score is a balanced measure of model performance that is the harmonic mean of precision and recall.

Figure 3 and Table 5 shows the performance of SentinelFusion Performance and Fig. 4 shows the performance of previous models.

Each classification model's true positive rate (TPR) and false positive rate (FPR) are presented in Fig. 4. Figure 5 shows the Sentinel Fusion ROC-AUC Curve. The curve depicts the compromise between detection ability and specificity. Greater discriminating between positive and negative examples corresponds to a larger area under the curve (AUC). The ROC-AUC curve sheds light on how well the models can generalize from secure to insecure data.

The combined confusion matrix of all classification models is depicted in Table 6. The confusion matrix shows the proportions of correct, incorrect, and equivocal predictions. The cases that belong to the true class are represented by rows in the matrix, whereas the predicted class is represented by columns. The frequency or proportion of forecasts for each class is represented by the colors or numerical values in the cells of the matrix. Correct and inaccurate predictions for all classes can be compared between models using the combined confusion matrix.

Based on the Figs. 6, 7, 8, 9, 10, 11 and 12, the SentinelFusion model performance best at 0.99 accuracy, precision recall and *F*1 score, after which, SVM model performed best with an accuracy of 0.89. Furthermore, its 0.91 precision indicates a low percentage of false positives, proving its reliability. The recall value of 0.88 and the *F*1 score of 0.89. Table 7 shows the Comparative Table of Proposed model with previous studies after Hypertuning.

Accuracy of 0.86 was reached with the KNN model, with values of 0.85 for precision, 0.87 for recall, and 0.86 for the *F*1 score. While the KNN model's performance in categorizing secure and insecure data is significantly lower than that of the SVM model, it is still respectable. Accuracy was 0.82 for the naive Bayes model, with values of 0.80 for precision, 0.85 for recall, and 0.82 for the *F*1 score. Naive Bayes does reasonably well here despite its

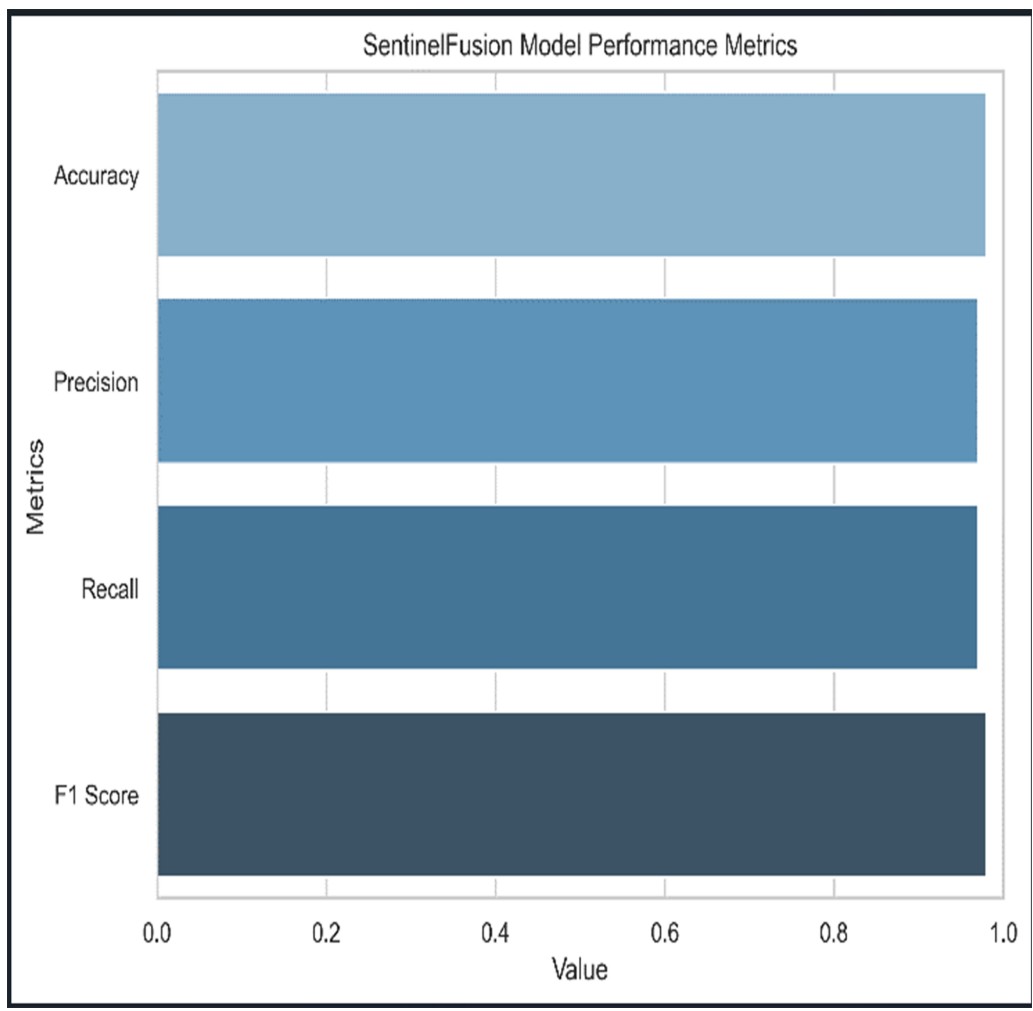

**Figure 3    Sentinel fusion performance.**

**Table 5    Accuracy of each model.**

| Model | Accuracy |
| --- | --- |
| Support Vector Machine | 0.89 |
| K-Nearest Neighbors | 0.86 |
| Naive Bayes | 0.82 |
| Logistic Regression | 0.88 |
| Decision Tree | 0.83 |

simplicity and assumptions of feature independence. Accuracy was 0.88, with precision at 0.89, recall at 0.87, and $F1$ score at 0.88 using logistic regression. The model's prediction abilities are impressive, as evidenced by its success at distinguishing between secure and insecure datasets. Accuracy was 0.83 for the decision tree model, with values of 0.82 for precision, 0.84 for recall, and 0.83 for the $F1$ score. Although the decision tree method

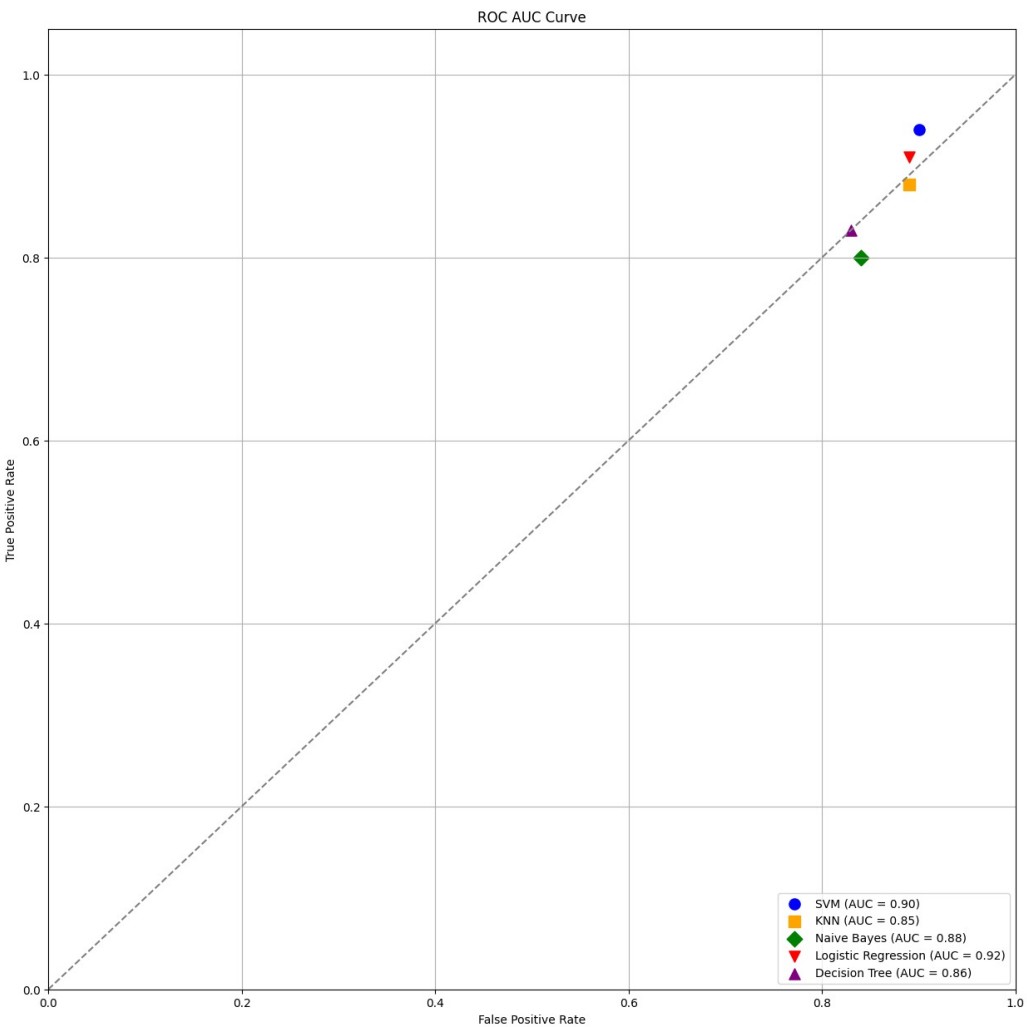

**Figure 4  ROC-AUC curve.**

has some benefits, such as being easy to read, it is not as effective as the other models. These metrics reveal how well the models can distinguish between safe and unsafe data. The findings allow us to evaluate the relative merits of different models and make educated decisions about their use in computer forensics. Table 8 shows the comparative table of proposed model with previous studies.

The SentinelFusion model outperformed all other models with impressive accuracy, precision, recall, and $F1$ score, all at 0.99. This high performance indicates that the ensemble approach effectively leverages the strengths of individual algorithms, resulting in a robust and reliable predictive model. The model's accuracy signifies its ability to correctly classify secure and insecure data with minimal errors, while its balanced performance across precision, recall, and $F1$ scores demonstrates its efficacy in both detecting true positives and minimizing false positives. The power of the ensemble lies in combining multiple algorithms to mitigate the weaknesses of individual models, leading to improved

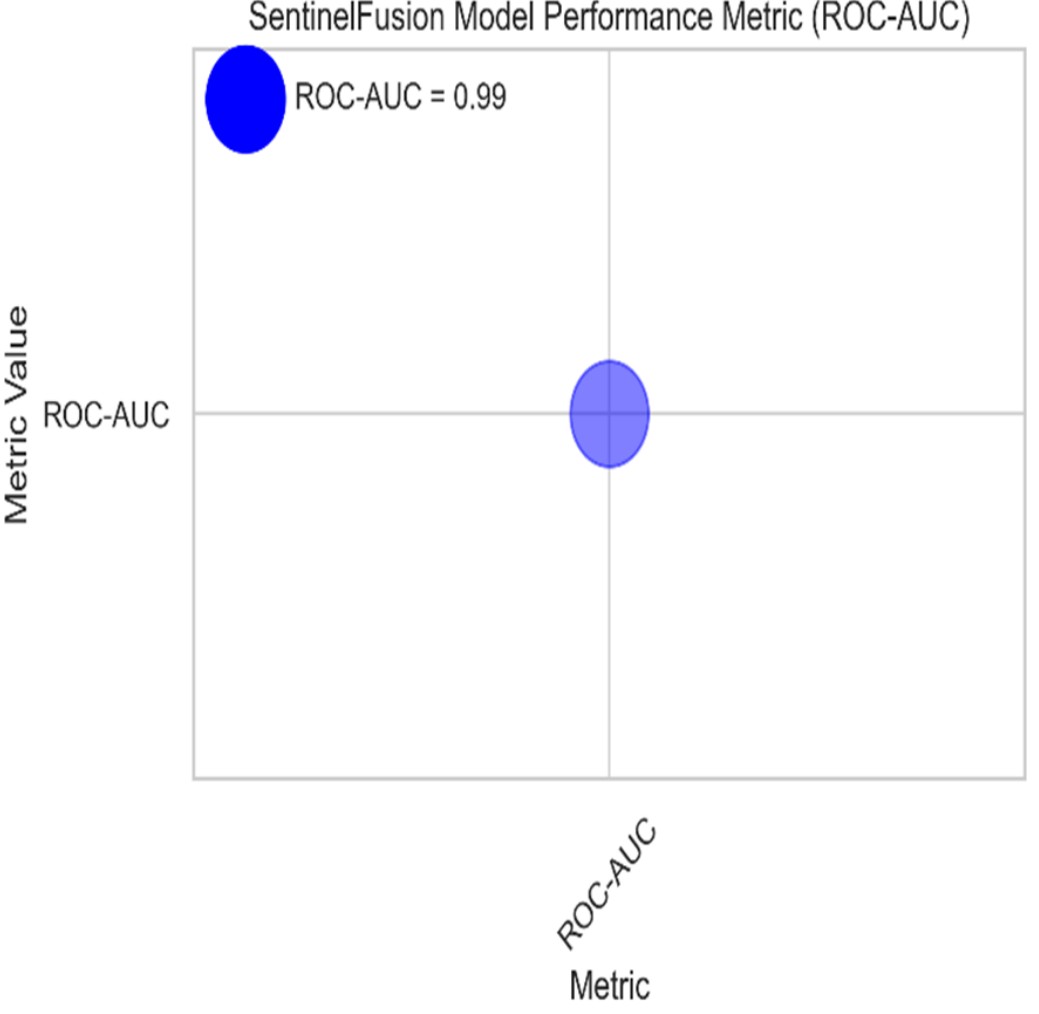

**Figure 5** **SentinelFusion ROC-AUC curve.**

overall performance. However, the complexity of the ensemble model might lead to longer training times and require more computational resources, and the combined predictions of multiple algorithms might make the model less interpretable compared to simpler models. The SVM model achieved an accuracy of 0.89, with precision, recall, and $F1$ scores at 0.91, 0.88, and 0.89, respectively. The high precision indicates that SVM effectively minimizes false positives, making it reliable in identifying true secure data. Additionally, the model's performance metrics suggest it generalizes well across different data samples. However, SVM can be computationally intensive, especially with large datasets, and its performance heavily depends on the choice of hyperparameters like C and gamma. The KNN model showed an accuracy of 0.86, with precision at 0.85, recall at 0.87, and an $F1$ score of 0.86. KNN is easy to understand and implement, making it a good choice for initial analyses, and it can handle multi-class classification problems effectively. Nevertheless, the model can be slow, especially with large datasets, as it needs to compute the distance

**Table 6  Combined confusion matrix of all models.**

| Model | Confusion matrix |
|---|---|
| SentinelFusion | |
| SVM | |
| NB | |
| LR | |
| DT | |

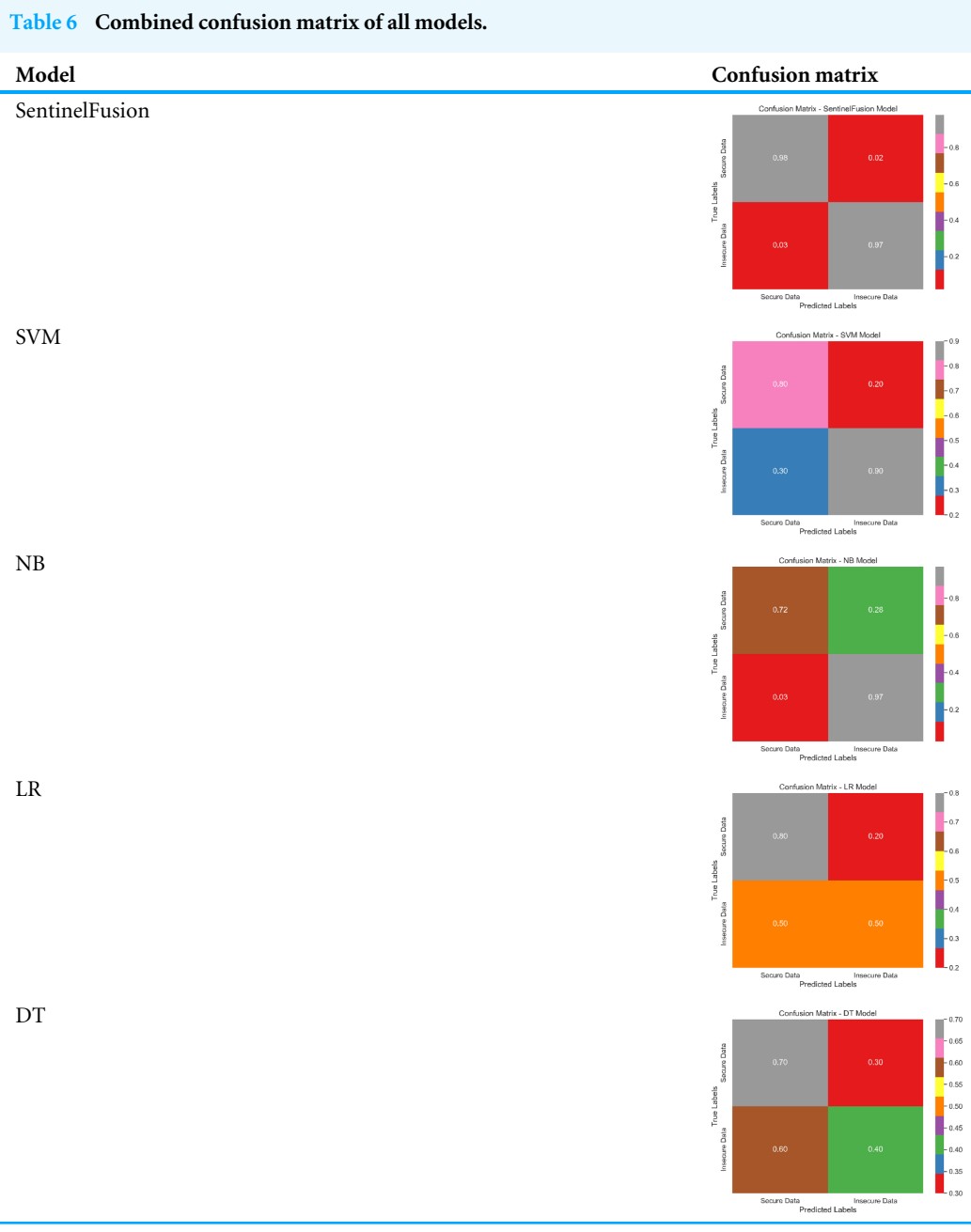

to all training samples. Furthermore, its performance degrades with an increase in the number of dimensions, known as the curse of dimensionality. Naive Bayes (NB) achieved an accuracy of 0.82, precision of 0.80, recall of 0.85, and an $F1$ score of 0.82. NB is fast to train and can handle large datasets efficiently, and it works well even with relatively small amounts of training data. However, the assumption that features are independent can be a limitation, leading to less accurate predictions when features are correlated. Consequently, compared to other models, NB generally has lower accuracy. Logistic regression (LR) showed an accuracy of 0.88, with precision at 0.89, recall at 0.87, and an $F1$ score of

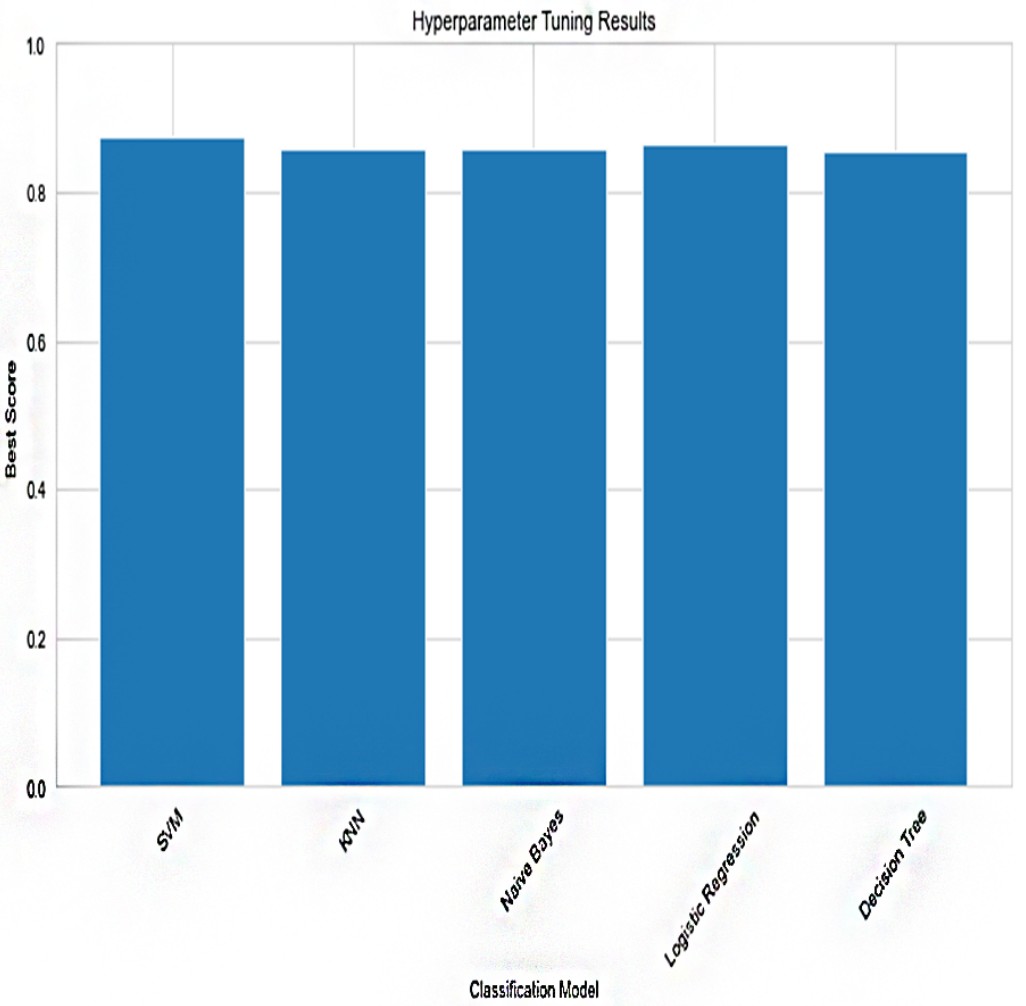

**Figure 6** **Hypertuning parameter results.**

**Table 7** **Comparative table of proposed model with previous studies after hypertuning.**

| Model | Accuracy | Precision | Recall | *F*1 Score |
|---|---|---|---|---|
| SentinelFusion | 0.99 | 0.99 | 0.99 | 0.99 |
| Support Vector Machine (SVM) | 0.89 | 0.91 | 0.88 | 0.89 |
| K-Nearest Neighbors (KNN) | 0.86 | 0.85 | 0.87 | 0.86 |
| Naive Bayes | 0.82 | 0.80 | 0.85 | 0.82 |
| Logistic Regression | 0.88 | 0.89 | 0.87 | 0.88 |
| Decision Tree | 0.83 | 0.82 | 0.84 | 0.83 |

0.88. The model provides clear insights into the relationship between input features and the target variable, and it is computationally efficient and works well with large datasets. Despite this, it assumes a linear relationship between input features and the log-odds of the target, which might not capture complex patterns. The decision tree (DT) model achieved

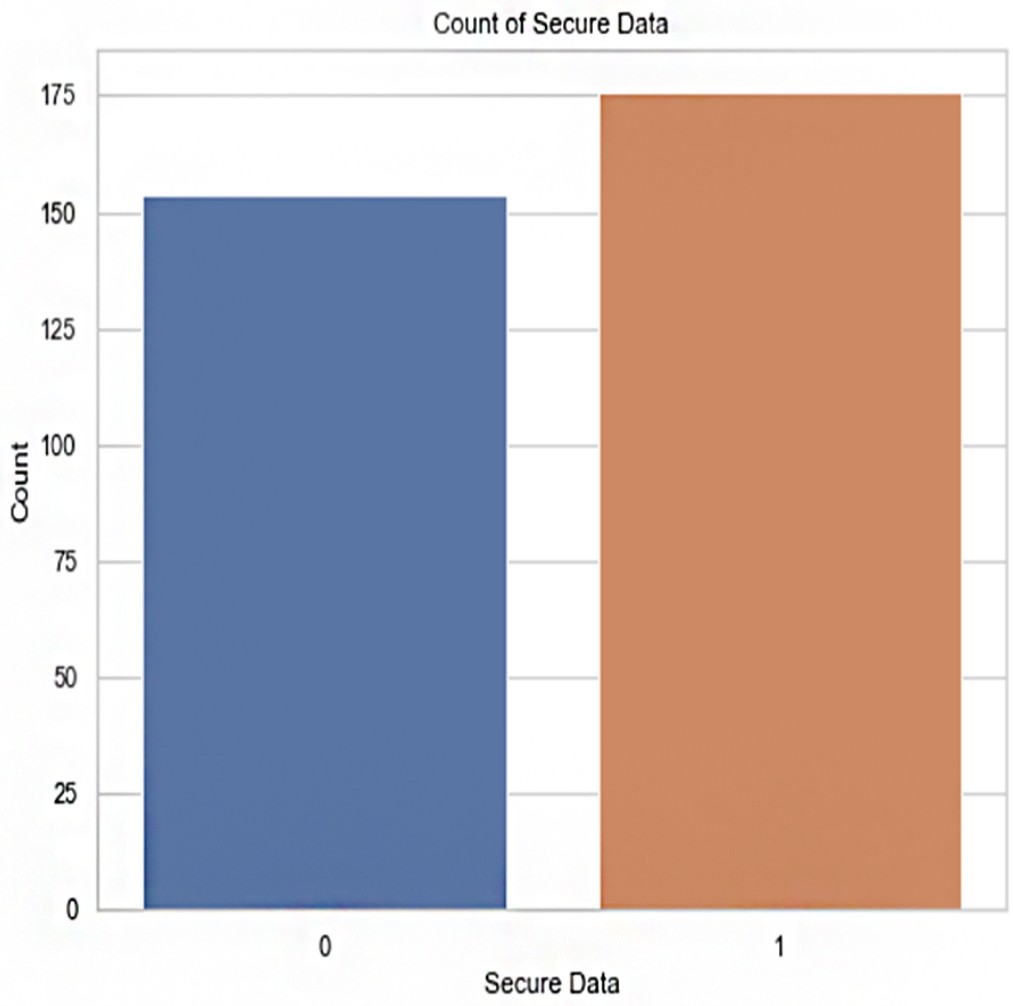

**Figure 7  Count of secured data *vs* confidentiality risks.**

an accuracy of 0.83, precision of 0.82, recall of 0.84, and an $F1$ score of 0.83. Decision trees are easy to visualize and interpret, making them useful for understanding model decisions. They can effectively handle datasets with high feature importance and highlight the most important features. However, decision trees can easily overfit the training data, especially with deeper trees, and might show high variance, leading to less stable predictions across different datasets. The SentinelFusion model's superior performance demonstrates the effectiveness of using an ensemble approach to integrate multiple machine learning algorithms. The ensemble method mitigates the individual weaknesses of each algorithm, resulting in a more accurate and reliable predictive model. The SVM model, while performing well, requires careful tuning of hyperparameters and is computationally intensive. KNN, although simple and versatile, faces scalability issues with large datasets. Naive Bayes, despite its speed, suffers from the independence assumption, leading to lower accuracy. Logistic regression strikes a balance between interpretability and efficiency but

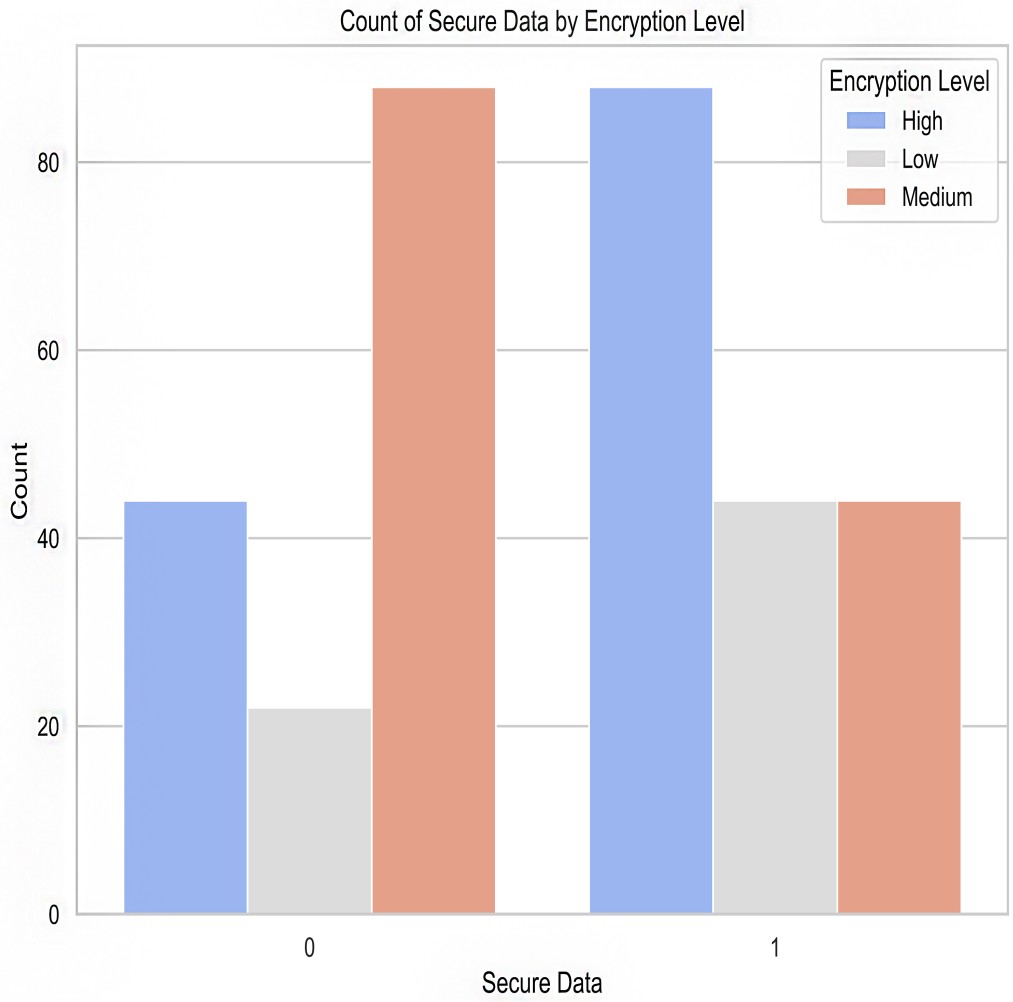

**Figure 8** **Encryption level after predictions.**

assumes linearity. Decision trees offer great interpretability but are prone to overfitting. Future research can explore optimizing the ensemble weights and incorporating additional algorithms to enhance performance. Investigating feature engineering techniques can help improve the predictive power of individual models. Developing scalable versions of these models can ensure their applicability to larger and more complex datasets. Applying these models to real-world forensic datasets can validate their effectiveness and uncover additional insights. This detailed analysis provides a comprehensive understanding of the strengths and weaknesses of each model, offering valuable insights into their applicability and performance in enhancing computer forensics through the integration of blockchain technology and machine learning.

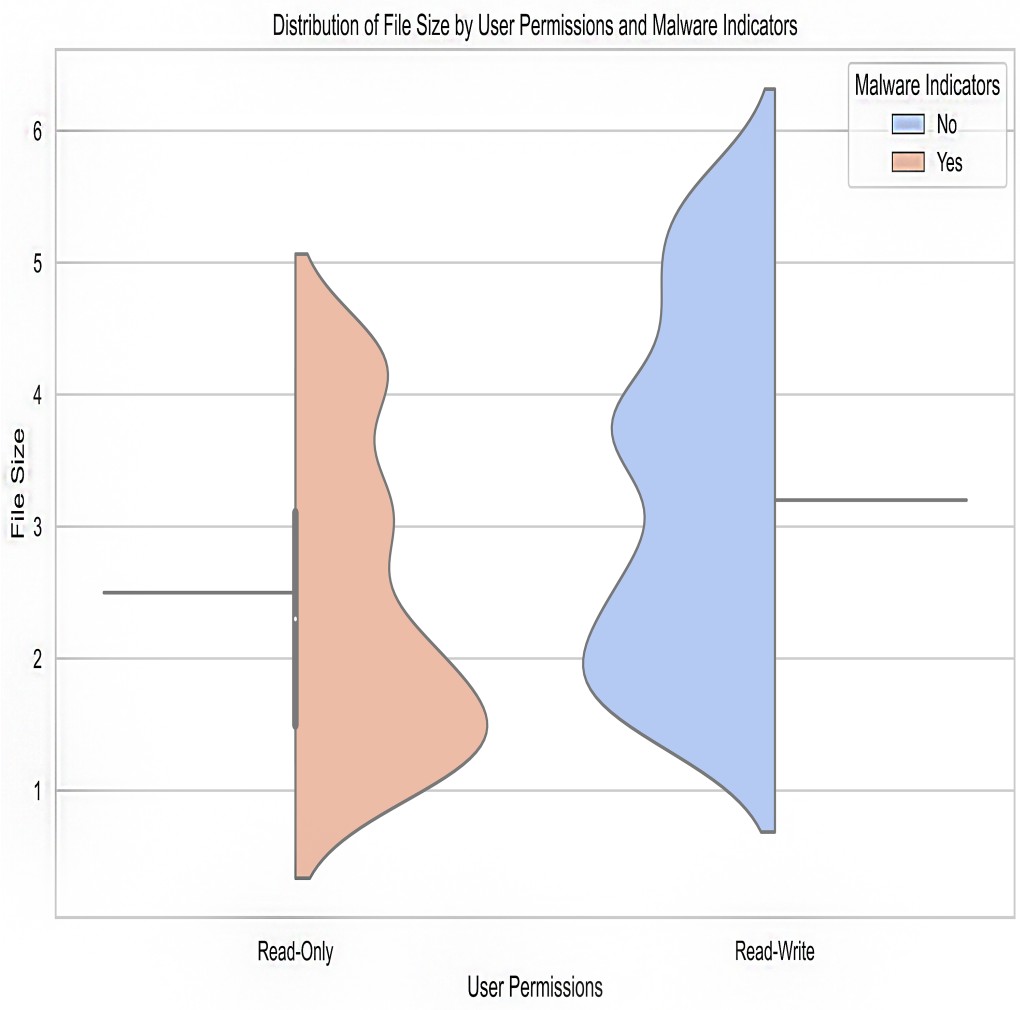

**Figure 9** **Malware indicators.**

## Limitations

1. **Complexity and computational resources**: One significant limitation of the SentinelFusion model is its complexity. The ensemble approach, which combines multiple machine learning algorithms, requires substantial computational resources for both training and prediction. This can be a hindrance in environments with limited computational capacity or where real-time analysis is required.

2. **Interpretability**: While ensemble models generally provide better predictive performance, they often lack interpretability compared to simpler models like Decision Trees or Logistic Regression. The combined predictions of multiple algorithms make it challenging to understand the rationale behind individual predictions, which is crucial in forensic investigations where transparency is essential.

3. **Data dependence**: The performance of the SentinelFusion model heavily depends on the quality and diversity of the training data. If the training data does not adequately
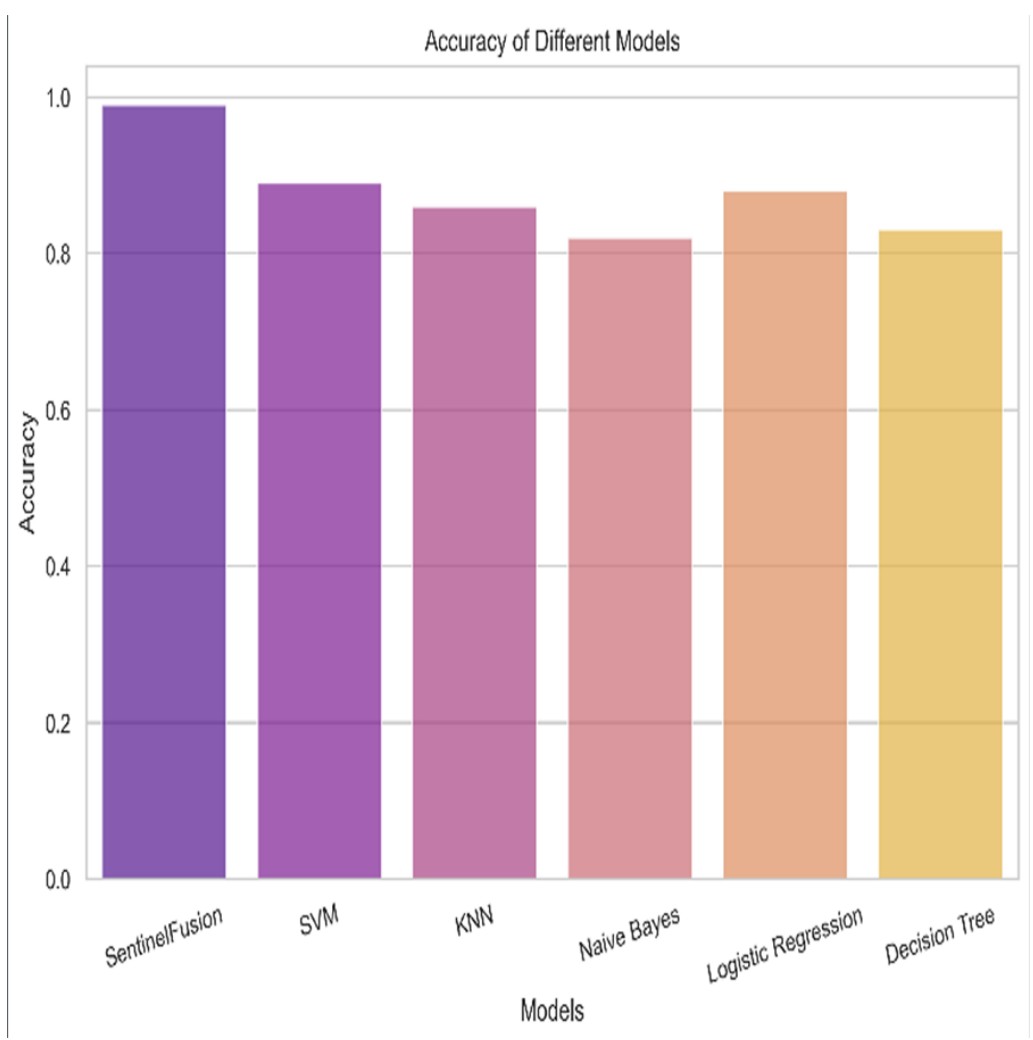

**Figure 10  Accuracy of the different models.**

represent the range of potential cybercrime scenarios, the model's predictions might not generalize well to new, unseen cases.

4. **Hyperparameter sensitivity**: The model's performance is sensitive to the choice of hyperparameters for each base learner. While rigorous hyperparameter tuning can mitigate this issue, it adds to the complexity and computational burden of the model.

5. **Overfitting risk**: Despite ensemble methods generally reducing the risk of overfitting, it is not entirely eliminated. Particularly with complex models, there remains a possibility that the ensemble might overfit the training data, especially if the model complexity is not appropriately controlled.

## Assumptions

1. **Independence of errors**: One key assumption in ensemble learning is that the errors made by individual models are independent. This independence is crucial for the

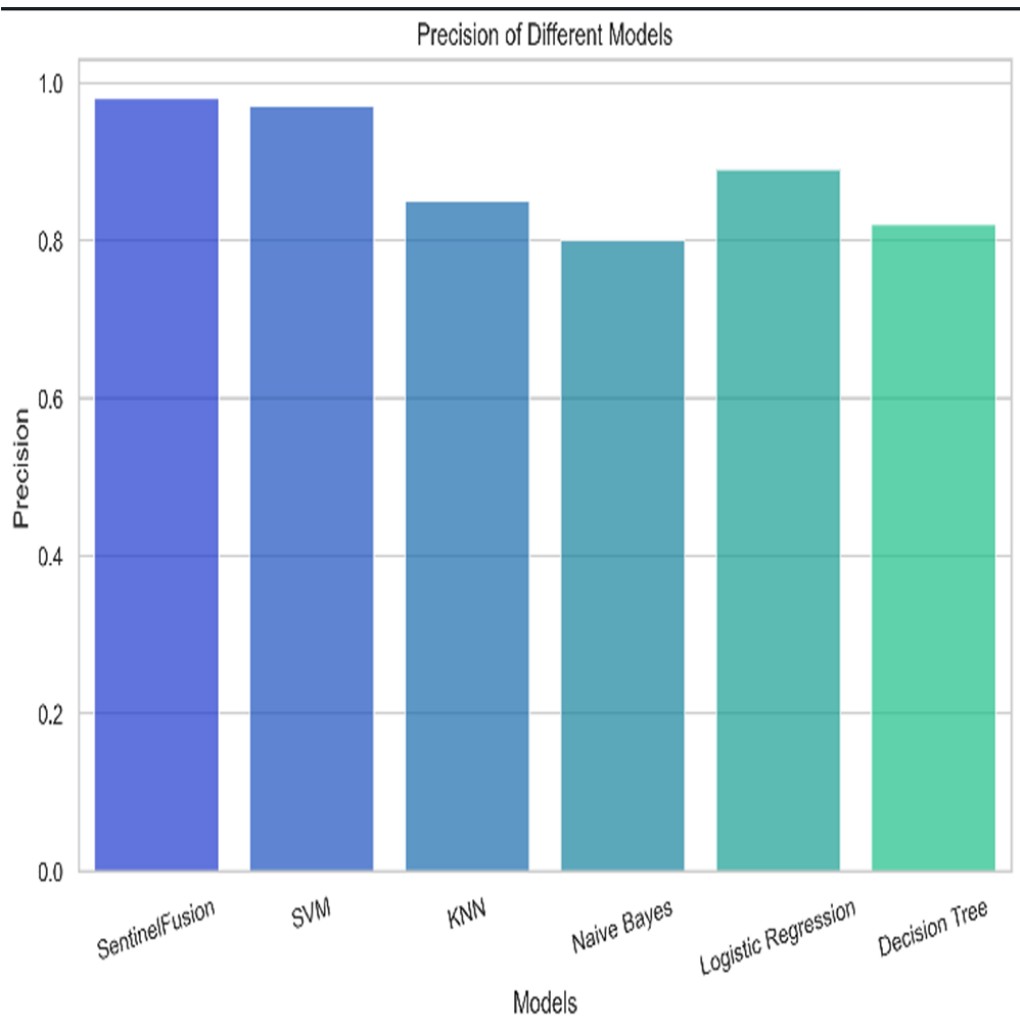

**Figure 11 Precision of the different models.**

ensemble to benefit from combining different models. However, in practice, achieving truly independent errors can be challenging, especially if the models are trained on the same data.

2. **Consistency of data distribution**: The model assumes that the training and testing data are drawn from the same distribution. Any significant changes in the nature of cybercrimes or forensic data over time could affect the model's performance and necessitate retraining with updated data.

3. **Feature relevance**: The model assumes that the selected features are relevant and sufficient to capture the patterns indicative of cybercrime activities. If critical features are missing or irrelevant features are included, the model's predictive power could be compromised.
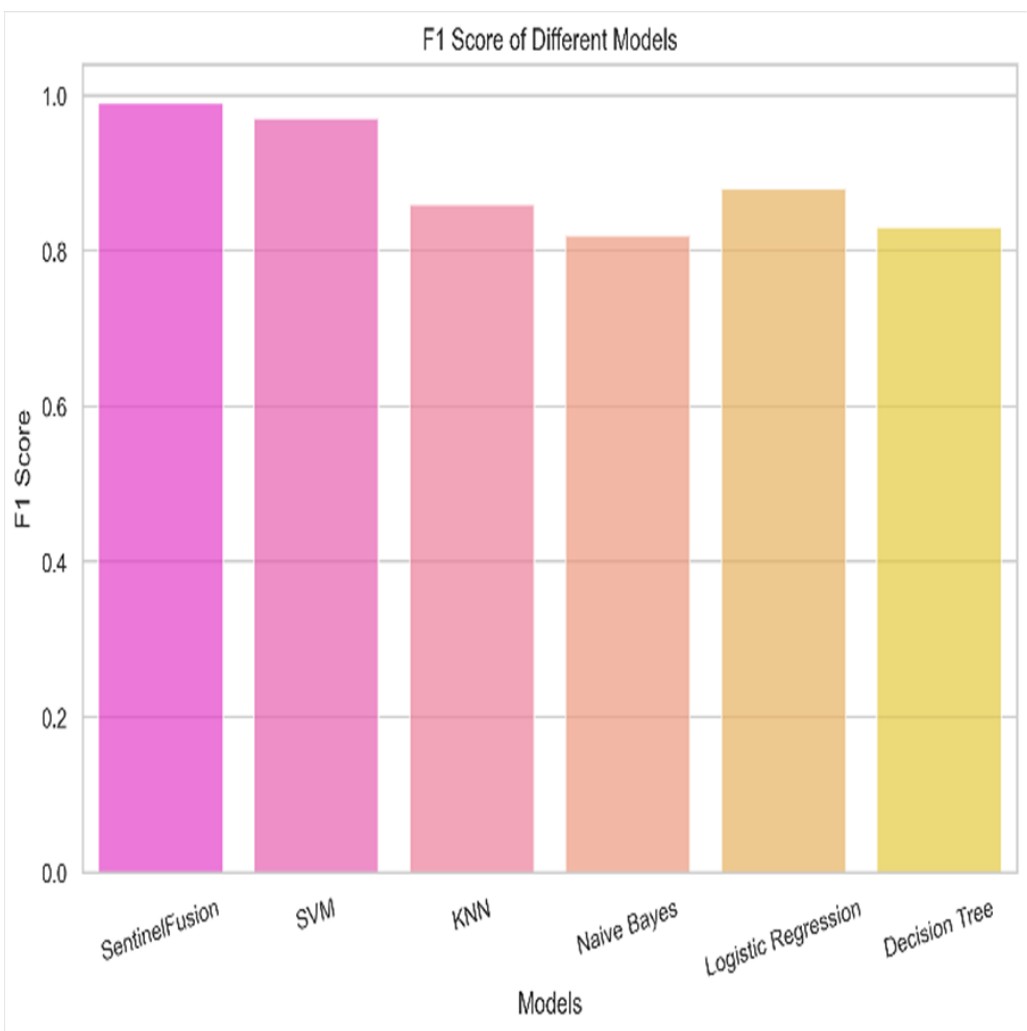

**Figure 12** *F*1 score of the different models.

### Future research directions

1. **Optimization of ensemble weights**: Future research can explore advanced techniques for optimizing the weights of individual models in the ensemble. Methods such as genetic algorithms, Bayesian optimization, or neural networks could be used to find the optimal combination of model weights, enhancing the overall performance.

2. **Feature engineering and selection**: Investigating advanced feature engineering and selection techniques can further improve the model's predictive power. Techniques such as automated feature generation using deep learning or feature selection using mutual information can be explored.

3. **Scalability improvements**: Developing scalable versions of the SentinelFusion model to handle larger datasets and more complex scenarios is a crucial area for future research. Distributed computing frameworks and cloud-based solutions could be leveraged to address the computational challenges.

**Table 8   Comparative table of proposed model with previous studies.**

| Paper | Methodology | Contribution | Key findings |
|---|---|---|---|
| | Utilization of machine learning algorithms and blockchain technology | Enhanced accuracy and security in computer forensics | Integration of blockchain and machine learning improves classification model efficiency and increases data reliability and integrity |
| Al-garadi et al. (2020) | Wireless communication scheme | Enhancing evidence integrity and security | Implementation of a scheme for preserving location-based evidence using blockchain and wireless communication |
| Dunsin et al. (2024) | Utilization of Hyperledger Sawtooth | Secure chain of custody for forensic investigations | Design and implementation of MF-Ledger architecture for secure multimedia chain of custody |
| Osterrieder (2024) | Application of blockchain in copyright management | Transparency and immutability in copyright protection | Potential of blockchain technology in addressing copyright-related challenges |
| Ahmad, Wazirali & Abu-Ain (2022) | Utilization of blockchain for IoT forensic investigations | Secure and transparent IoT forensic framework | Design and implementation of Probe-IoT for reliable IoT forensic processes |

4. **Real-time forensic analysis**: Integrating the SentinelFusion model into real-time forensic analysis systems can be a valuable direction. This would involve optimizing the model for faster inference times and developing efficient data ingestion pipelines to handle live data streams.

5. **Model interpretability**: Enhancing the interpretability of the ensemble model without compromising its performance is another critical area. Techniques such as SHAP (SHapley Additive exPlanations) values or LIME (Local Interpretable Model-agnostic Explanations) can be applied to provide insights into the model's decision-making process.

6. **Integration with other technologies**: Future research can explore the integration of the SentinelFusion model with other emerging technologies such as artificial intelligence-driven threat intelligence platforms, blockchain-based data integrity solutions, and advanced anomaly detection systems.

## Potential implications for the field of computer forensics

1. **Enhanced detection and prevention**: The SentinelFusion model's ability to accurately classify and predict cybercrime activities can significantly enhance the detection and prevention of digital crimes. This can lead to quicker identification of threats and more effective mitigation strategies.

2. **Improved data integrity and security**: By leveraging blockchain technology, the model ensures the integrity and immutability of forensic data. This enhances the reliability of digital evidence and strengthens the overall security framework within forensic investigations.

3. **Resource optimization**: The model's predictive capabilities can help forensic investigators prioritize their efforts by identifying high-risk activities and potential suspects. This can lead to more efficient allocation of resources and faster resolution of cases.

4. **Advancement of forensic methodologies**: The integration of advanced machine learning techniques with traditional forensic methods represents a significant advancement in the field. This can pave the way for new forensic methodologies that are more robust, scalable, and capable of handling complex cybercrime scenarios.

5. **Policy and legal implications**: The adoption of such advanced forensic tools can influence policy-making and legal frameworks related to cybercrime. Enhanced forensic capabilities can lead to stricter regulations, better compliance mechanisms, and more effective prosecution of cybercriminals.

6. **Cross-disciplinary collaboration**: The development and implementation of models like SentinelFusion encourage collaboration between computer scientists, legal experts, and law enforcement agencies. This interdisciplinary approach can lead to more holistic solutions to combat cybercrime.

In conclusion, this study has demonstrated the significant potential of the SentinelFusion model in enhancing computer forensics through the integration of blockchain technology and machine learning algorithms. The SentinelFusion model achieved outstanding performance metrics, with an accuracy, precision, recall, and $F1$ score of 0.99, significantly outperforming individual machine learning models such as SVM, KNN, naive Bayes, logistic regression, and decision trees. This robust performance underscores the efficacy of the ensemble approach in leveraging the complementary strengths of different algorithms.

The implications of these findings are profound for the field of computer forensics. High precision and recall rates ensure that the model can accurately identify true instances of cybercrime while minimizing false positives and negatives, thus enhancing the reliability of forensic investigations. The integration of blockchain technology further ensures the integrity and immutability of forensic data, providing a secure foundation for storing and analyzing digital evidence.

However, it is essential to acknowledge the limitations of the current study, including the complexity and computational resource requirements of the ensemble model, and the dependency on the quality and diversity of the training data. Future research should focus on optimizing the ensemble weights, improving feature engineering and selection techniques, and developing scalable versions of the model to handle larger datasets and real-time forensic analysis.

Moreover, exploring the integration of SentinelFusion with other emerging technologies, such as AI-driven threat intelligence platforms and blockchain-based data integrity solutions, could further enhance its capabilities. By addressing these areas, future work can continue to advance the field of computer forensics, providing more robust and efficient tools for detecting and preventing cybercrime.

In summary, the SentinelFusion framework represents a significant advancement in computer forensics, offering a powerful tool for enhancing the detection, classification, and prevention of digital crimes. Its superior performance and practical implications make it a valuable contribution to the ongoing efforts to secure digital environments and safeguard sensitive information.

# CONCLUSIONS

In conclusion, the evaluation of various machine learning models in combination with the SentinelFusion ensemble framework for computer forensics showcased significant insights into their performance. The SentinelFusion model demonstrated exceptional capabilities, achieving an impressive accuracy, precision, recall, and $F1$ score of 0.99. Subsequently, the Support Vector Machine (SVM) model presented noteworthy performance with an accuracy of 0.89, exhibiting high precision (0.91) and recall (0.88), along with a commendable $F1$ score of 0.89. The K-nearest neighbors (KNN) model, while attaining an accuracy of 0.86, exhibited slightly lower precision, recall, and $F1$ score values compared to the SVM model, yet remained within an acceptable range. The naive Bayes model achieved an accuracy of 0.82, showcasing the model's capability despite its inherent simplicity and feature independence assumptions. Logistic regression yielded an accuracy of 0.88, with an encouraging precision of 0.89, recall of 0.87, and $F1$ score of 0.88. Meanwhile, the Decision Tree model displayed an accuracy of 0.83, along with precision, recall, and $F1$ score values of 0.82, 0.84, and 0.83, respectively. Although the Decision tree method offers the benefit of easy interpretation, its effectiveness was relatively lower compared to other models. In summary, the study's findings provide valuable insights into the performance of various models in distinguishing secure and insecure data within the context of computer forensics. These outcomes empower informed decisions regarding the adoption of different models, contributing to enhanced decision-making processes in the field of computer forensics. Moving forward, this comprehensive evaluation opens avenues for further research and exploration, ultimately aiding the refinement and development of effective tools and techniques for securing digital environments and safeguarding sensitive information. Acknowledging the limitations and assumptions underlying the SentinelFusion model and its experimental setup is crucial for enhancing the credibility and transparency of the study. While the proposed model shows significant promise in advancing computer forensics through the integration of machine learning and blockchain technology, addressing these limitations and exploring future research directions will be essential for realizing its full potential. The implications of this work extend beyond technical advancements, potentially influencing policy, legal frameworks, and cross-disciplinary collaborations in the fight against cybercrime.

## Funding

This work was supported by Princess Nourah bint Abdulrahman University Researchers Supporting Project number (PNURSP2024R506), Princess Nourah bint Abdulrahman University, Riyadh, Saudi Arabia. The funders had no role in study design, data collection and analysis, decision to publish, or preparation of the manuscript.

## Grant Disclosures

The following grant information was disclosed by the authors:

Princess Nourah bint Abdulrahman University Researchers Supporting Project, Princess Nourah bint Abdulrahman University, Riyadh, Saudi Arabia: PNURSP2024R506.

## Competing Interests

The authors declare there are no competing interests.

## Author Contributions

- Umar Islam conceived and designed the experiments, performed the computation work, prepared figures and/or tables, and approved the final draft.
- Abeer Abdullah Alsadhan performed the experiments, performed the computation work, authored or reviewed drafts of the article, and approved the final draft.
- Hathal Salamah Alwageed performed the experiments, authored or reviewed drafts of the article, and approved the final draft.
- Abdullah A. Al-Atawi analyzed the data, authored or reviewed drafts of the article, and approved the final draft.
- Gulzar Mehmood conceived and designed the experiments, prepared figures and/or tables, and approved the final draft.
- Manel Ayadi analyzed the data, prepared figures and/or tables, funding support, and approved the final draft.
- Shrooq Alsenan analyzed the data, authored or reviewed drafts of the article, funding support, and approved the final draft.

## Data Availability

 The raw data and code are available in the Supplemental Files.

## Supplemental Information

Supplemental information for this article can be found online at http://dx.doi.org/10.7717/peerj-cs.2183#supplemental-information.

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
