# Peer review of "SentinelFusion based machine learning comprehensive approach for enhanced computer forensics"

_PeerJ Computer Science, doi:10.7717/peerj-cs.2183_

## Round 0.1 · original submission · Major Revisions

More details of comparative analysis are required.

·

Basic reporting

The presentation of results through figures and tables is commendable, but there is a need for more detailed interpretation and analysis. The discussion should delve deeper into the patterns and trends observed in the results, providing insights into the strengths and weaknesses of each model.

Experimental design

Comment 1: Provide the conceptual framework that underpins SentinelFusion. A more detailed introduction to the theoretical foundations of ensemble learning within the realm of computer forensics is required to afford readers a comprehensive understanding of the proposed model.

Comment 2: The comparative analysis between SentinelFusion and baseline models is limited in scope and depth. A more comprehensive comparison with existing state-of-the-art ensemble frameworks in the field of computer forensics is necessary to contextualize the novelty and effectiveness of the proposed approach.

Validity of the findings

Comment 3: The manuscript overlooks the discussion of potential limitations and assumptions underlying the proposed model and experimental setup. Acknowledging and addressing these factors, along with their implications for the validity and generalizability of the results, is crucial to enhance the credibility and transparency of the study.
Discus the future research directions and potential implications for the field of computer forensics.

Additional comments

A proofread is required.

Reviewer 2 ·

Basic reporting

The level of English should be improved in the revised version.The manuscript lacks clear articulation of the conceptual framework underlying SentinelFusion. A more elaborate introduction to the theoretical foundations of ensemble learning in the context of computer forensics is necessary to provide readers with a comprehensive understanding of the proposed model.

The description of SentinelFusion's architecture is superficial and lacks specificity regarding the integration of individual machine-learning algorithms. Detailed illustrations and diagrams elucidating the flow of information and the decision-making process within the ensemble framework would greatly enhance clarity and facilitate comprehension.

While the ensemble approach is justified in principle, the manuscript fails to provide a robust rationale for the specific combination of machine learning algorithms employed in SentinelFusion. A more rigorous analysis of the complementary strengths and weaknesses of the constituent models is essential to support the efficacy of the ensemble framework.

Experimental design

The experimental design lacks methodological rigor, particularly in terms of dataset selection, partitioning strategy, and cross-validation protocol. Providing detailed explanations of these aspects, along with justification for the chosen methodology, is imperative to ensure the validity and reliability of the study's findings.
The optimization of hyperparameters is briefly mentioned but lacks comprehensive detail regarding the specific techniques employed and the rationale behind their selection. A more systematic approach to hyperparameter tuning, coupled with sensitivity analysis and validation procedures, is necessary to optimize model performance effectively.
Evaluation Metrics and Interpretation: The interpretation of evaluation metrics is cursory and lacks depth, particularly in terms of discussing the practical implications of model performance on real-world applications. A more nuanced analysis of metrics such as precision, recall, and F1 score, along with their implications for forensic analysis, is essential to derive meaningful insights from the results.

Validity of the findings

An explanation of the comparison of the proposed approach with existing methods is required.
Incorporate relevant references to recent studies in this area to provide a comprehensive review of the existing literature.
Provide recommendations for future research directions.
The conclusion section should be modified by adding key findings.

Reviewer 3 ·

Basic reporting

The article, titled “SentinelFusion based machine learning comprehensive approach for enhanced computer forensics” provide the potential enhancement of secrecy, privacy, and data integrity within blockchain systems by amalgamating the innate security features of blockchain with the predictive abilities of machine learning. Following are some of my major comments.
1. Starting from the abstract, the abstract does not correlate with the content of the article. Please rewrite the abstract and try to make it more informative. Besides, the lines in the conclusion section are not well synchronized and interlinked, I would suggest revising it.
2. I would suggest to provide the conceptual framework that underpins SentinelFusion. A more detailed introduction to the theoretical foundations of ensemble learning within the realm of computer forensics is required to afford readers a comprehensive understanding of the proposed model.
3. The comparative analysis between SentinelFusion and baseline models is limited in scope and depth. A more comprehensive comparison with existing state-of-the-art ensemble frameworks in the field of computer forensics is necessary to contextualize the novelty and effectiveness of the proposed approach.
4. The manuscript overlooks the discussion of potential limitations and assumptions underlying the proposed model and experimental setup. Acknowledging and addressing these factors, along with their implications for the validity and generalizability of the results, is crucial to enhance the credibility and transparency of the study.
5. The author does not mention future direction of the specific method. Discus the future research directions and potential implications for the field of computer forensics.
6. The presentation of results through figures and tables is commendable, but there is a need for more detailed interpretation and analysis. The discussion should delve deeper into the patterns and trends observed in the results, providing insights into the strengths and weaknesses of each model.
7. There are many typos and grammatical mistakes in the paper, which need to correct before further proceeding.

Experimental design

mentioned above

Validity of the findings

mentioned above

Additional comments

mentioned above

---

## Round 0.2 · accepted · Accept

Authors addressed all of mine and reviewers comments. The paper is now acceptable for publication.

·

Basic reporting

Significant enhancements

Experimental design

Everything is Clear, well experimented and defined.

Validity of the findings

No comments

Additional comments

I recommend the paper for publication in it's current form.

Reviewer 2 ·

Basic reporting

I have checked the revised version. The authors addressed all comments. Thanks

Experimental design

The changes are satisfactory and the manuscript is acceptable.

Validity of the findings

The revised version of manuscript is acceptable.